# A Meier-Gorlin syndrome mutation in a conserved C-terminal helix of Orc6 impedes origin recognition complex formation

**Franziska Bleichert[1,2†], Maxim Balasov[3], Igor Chesnokov[3], Eva Nogales[2,4,5], Michael R Botchan[2]\*, James M Berger[2]\*†**

[1]Miller Institute for Basic Research in Science, University of California, Berkeley, Berkeley, United States; [2]Department of Molecular and Cell Biology, University of California, Berkeley, Berkeley, United States; [3]Department of Biochemistry and Molecular Genetics, University of Alabama at Birmingham School of Medicine, Birmingham, United States; [4]Life Sciences Division, Lawrence Berkeley National Laboratory, Berkeley, United States; [5]Howard Hughes Medical Institute, University of California, Berkeley, Berkeley, United States

**Abstract** In eukaryotes, DNA replication requires the origin recognition complex (ORC), a six-subunit assembly that promotes replisome formation on chromosomal origins. Despite extant homology between certain subunits, the degree of structural and organizational overlap between budding yeast and metazoan ORC has been unclear. Using 3D electron microscopy, we determined the subunit organization of metazoan ORC, revealing that it adopts a global architecture very similar to the budding yeast complex. Bioinformatic analysis extends this conservation to Orc6, a subunit of somewhat enigmatic function. Unexpectedly, a mutation in the Orc6 C-terminus linked to Meier-Gorlin syndrome, a dwarfism disorder, impedes proper recruitment of Orc6 into ORC; biochemical studies reveal that this region of Orc6 associates with a previously uncharacterized domain of Orc3 and is required for ORC function and MCM2–7 loading in vivo. Together, our results suggest that Meier-Gorlin syndrome mutations in Orc6 impair the formation of ORC hexamers, interfering with appropriate ORC functions.

**\*For correspondence:**
mbotchan@berkeley.edu (MRB);
jmberger@jhmi.edu (JMB)

†**Present address:** Department of Biophysics and Biophysical Chemistry, Johns Hopkins University School of Medicine, Baltimore, United States

**Reviewing editor**: Johannes Walter, Harvard Medical School, United States

## Introduction

The faithful inheritance of genetic information depends on the precise replication of DNA; errors such as under- or over-replication of DNA can lead to genetic instabilities, malignant transformation, or cell death (*Klingseisen and Jackson, 2011*; *Abbas et al., 2013*). In eukaryotes, replication initiation machineries are loaded onto DNA during the G1-phase of the cell cycle, 'licensing' origins of replication for subsequent replisome assembly in S-phase (*Masai et al., 2010*; *Abbas et al., 2013*). The eukaryotic initiator, ORC (origin recognition complex), performs a central role in this licensing reaction (*Bell and Stillman, 1992*), working with the loading factors Cdc6 and Cdt1 to recruit the MCM2–7 (minichromosome maintenance) replicative helicase into a pre-replicative complex (pre-RC) at replication origins (*Liang et al., 1995*; *Cocker et al., 1996*; *Tanaka et al., 1997*; *Maiorano et al., 2000*; *Nishitani et al., 2000*). The MCM complex is loaded onto double-stranded DNA as an inactive double-hexamer (*Evrin et al., 2009*; *Remus et al., 2009*; *Gambus et al., 2011*), and is activated only after origin firing in the subsequent S-phase to commence DNA replication (*Vijayraghavan and Schwacha, 2012*). How ORC accurately executes the replication initiation program, and how disregulation of ORC function is linked to chromosomal abnormalities and disease, constitute long-standing questions in the field.

**eLife digest** Cell division is essential for organisms to be able to grow, to repair tissues and to proliferate. However, cells can only divide once they have successfully replicated their DNA. Many different molecules are involved in these two processes, including a large multi-protein assembly called the origin recognition complex that helps to start the process of DNA replication.

This complex contains six proteins but relatively little is known about its structure. It is also unclear how much origin recognition complexes (ORCs) differ between species. Now, Bleichert et al. have found a way to stabilize a specific conformation of *Drosophila* ORC, and have gone on to determine its structure at a higher resolution than was previously possible.

This approach revealed that the arrangement of protein subunits in *Drosophila* ORC is similar to that found in yeast ORC. Most of the ORC subunits have similar amino acid sequences in both species. However, the ORC6 subunit was regarded a notable exception for a long time, with the yeast and *Drosophila* versions of this subunit having different sequences of amino acids. Bleichert et al. show that the ORC6 subunits actually have important similarities, both in sequence and in function. In particular, the C-terminus of the ORC6 protein contains similar amino acids in both yeast and *Drosophila*. Moreover, it performs the same role—binding to another subunit—in both yeast and *Drosophila*.

As well as being important for cell division, human ORC has been implicated in Meier-Gorlin syndrome, a type of dwarfism. Mutations in three of the six ORC subunits, including ORC6, have been found in people with Meier-Gorlin syndrome. The mutations in ORC6 that are associated with this syndrome are in the C-terminus, which suggests that some symptoms of the syndrome may be caused by DNA replication not being initiated correctly. Consistent with this idea, Bleichert et al. show that the introduction of the Meier-Gorlin syndrome mutation into ORC6 prevents this subunit from binding to the rest of ORC, and similar mutations do not support DNA replication in in vivo experiments. These results should increase our understanding of the function of ORC6 and its role in Meier-Gorlin syndrome, and also provide new insights into the changes in ORC architecture that have occurred during evolution.

ORC is a heterohexameric protein complex containing the subunits Orc1–Orc6 (*Bell and Stillman, 1992*; *Gavin et al., 1995*; *Gossen et al., 1995*; *Rowles et al., 1996*; *Moon et al., 1999*; *Vashee et al., 2001*). The Orc1–Orc5 subunits all contain AAA$^+$ (ATPases associated with a variety of cellular activities) or AAA$^+$-like domains (*Neuwald et al., 1999*; *Iyer et al., 2004*; *Speck et al., 2005*; *Clarey et al., 2006*), a subset of which use ATP binding and hydrolysis to support replicative helicase loading, DNA replication, and cell viability (*Chesnokov et al., 2001*; *Bowers et al., 2004*; *Giordano-Coltart et al., 2005*; *Randell et al., 2006*). How individual ORC subunits co-assemble with each other and the nature of the structural transitions that occur within ORC and other pre-RC components during nucleotide-dependent initiation steps are ill-defined, in part due to a paucity of structures for key ORC reaction states. Electron microscopy (EM) studies have provided medium-to-low resolution pictures of the general architecture of *Saccharomyces cerevisiae* and *Drosophila* ORC, and subunit positions within ORC further have been systematically mapped for the yeast complex (*Speck et al., 2005*; *Clarey et al., 2006*; *Chen et al., 2008*; *Sun et al., 2012*, *2013*); however, poor similarity between the 3D reconstructions obtained for the different species, combined with conflicting biochemical data on ORC subunit interactions, have led to different models of subunit arrangement and DNA engagement for the two eukaryotic initiators (*Vashee et al., 2001*; *Kneissl et al., 2003*; *Chen et al., 2007*; *Matsuda et al., 2007*; *Siddiqui and Stillman, 2007*; *Clarey et al., 2008*; *Li and Stillman, 2012*; *Sun et al., 2012*). Whether these differences reflect true differences in structure, or potentially distinct functional states, remains unclear.

Recently, mutations in genes encoding the pre-RC components Orc1, Orc4, Orc6, Cdc6, and Cdt1 have been linked to Meier-Gorlin syndrome (MGS), a rare genetic disorder characterized by primordial dwarfism, small ears and aplastic or hypoplastic patella (*Guernsey et al., 2011*; *Bicknell et al., 2011a, b*). Based on the fact that MGS mutations cluster in pre-RC components, it has been suggested that the clinical phenotype is caused by defects in DNA replication initiation (*Guernsey et al., 2011*; *Bicknell et al., 2011a, b*). Consistent with this interpretation, cells derived from MGS patients with mutations in Orc1 show reduced recruitment of ORC subunits and replicative helicase components to chromatin,

and are compromised in their ability to replicate plasmid DNA (*Bicknell et al., 2011b*); recent work has also shown a link between origin licensing factors and non-replicative cellular processes, such as centrosome duplication and cilia formation, in MGS patient cells and in cells depleted of licensing proteins (*Hossain and Stillman, 2012*; *Stiff et al., 2013*). Most MGS mutations in Orc1 lie in the N-terminal domain, which is not part of the central core of ORC, and have been implicated interfering with Orc1's Cyclin E-CDK2 inhibitory function of centrosome duplication (*Hossain and Stillman, 2012*). By contrast, the means by which certain mutations in Orc4 and Orc6 result in an MGS phenotype has yet to be determined. Orc6 is particularly enigmatic in this regard: unlike Orc1–Orc5, Orc6 is poorly conserved between budding yeast and metazoans, both in sequence and in aspects of its function (*Lee and Bell, 1997*; *Chesnokov et al., 2001*; *Prasanth et al., 2002*; *Chesnokov et al., 2003*; *Semple et al., 2006*; *Balasov et al., 2007*; *Chen et al., 2007*; *Duncker et al., 2009*; *Bernal and Venkitaraman, 2011*; *Chen and Bell, 2011*; *Liu et al., 2011*; *Takara and Bell, 2011*).

Here we investigate the structural organization of *Drosophila* ORC (*Dm*ORC) and how Orc6 is recruited into this complex in metazoans. Using EM we find that apo-*Dm*ORC is conformationally heterogeneous, but that the non-hydrolysable nucleotide analog ATPγS locks the complex in a specific conformation that permitted the determination of its 3D structure to 22 Å resolution. Experimental localization of all subunits reveals that the subunit architecture of *Dm*ORC is preserved compared to *S. cerevisiae* ORC (*Sc*ORC), reconciling different models proposed previously for the two systems. Interestingly, sequence analyses show that this similarity extends to the domain structure of Orc6, suggesting analogous functions for Orc6 from yeast and metazoans. Consistent with this notion, we find that recruitment of Orc6 into ORC across multiple species is aided by the interaction of its C-terminus with a novel domain that resides between the AAA+-like and winged-helix domains of Orc3. Strikingly, a mutation found in patients with Meier-Gorlin syndrome maps to a highly-conserved C-terminal segment in Orc6, destabilizing the interaction of *Drosophila* and human Orc6 with both Orc3 alone and with the core Orc1–5 subcomplex. Orc6 transgenes containing mutations in the conserved C-terminal tail fail to incorporate into *Dm*ORC in vivo, and do not complement an *orc6*-null allele when introduced into flies. Taken together, these results suggest that the MGS mutation in Orc6 directly affects the integrity of ORC, and indicate that destabilization of ORC contributes to the pathogenesis of Meier-Gorlin syndrome caused by mutations in Orc6, ultimately interfering with pre-RC assembly and origin licensing.

## Results

### Improved resolution EM structure of *Drosophila* ORC

EM reconstructions of *S. cerevisiae* ORC and *Drosophila* ORC have been determined previously (*Speck et al., 2005*; *Clarey et al., 2006*; *Sun et al., 2012*, *2013*), and have allowed individual subunits to be localized in budding yeast ORC. Unfortunately, the modest resolution of these structures, together with the poor resemblance between the *Drosophila* and *S. cerevisiae* ORC models, have made it difficult to extrapolate the yeast subunit order into metazoan ORC (*Speck et al., 2005*; *Clarey et al., 2006*, *2008*; *Chen et al., 2008*; *Sun et al., 2012*). To overcome this gap, we set out to improve the resolution of the *Drosophila* ORC model by negative-stain electron microscopy and to further map the complex's higher-order organization by subunit tagging and visualization.

To obtain samples for EM, we purified recombinant *Drosophila* ORC from insect cells (*Figure 1—figure supplement 1A*). As previously observed, *Drosophila* ORC forms a stable, hexameric complex containing subunits Orc1–Orc6 (*Figure 1A*, *Figure 1—figure supplement 1B*) (*Chesnokov et al., 1999*, *2001*). Examination of ORC by EM after negative staining showed a monodisperse population of particles, with no indication of complex instability at the low-nanomolar concentrations used for sample deposition onto EM grids (*Figure 1—figure supplement 1C*).

Nucleotide binding and hydrolysis often generate significant conformational changes in oligomeric assemblies of AAA+ ATPases (*Neuwald et al., 1999*; *Iyer et al., 2004*), a behavior seen previously by EM for *Drosophila* ORC in the absence and presence of the non-hydrolysable ATP analog ATPγS (*Clarey et al., 2006*). However, the maximal obtainable resolution for these early EM reconstructions was relatively low (34 Å and 33 Å resolution, respectively), suggesting that conformational heterogeneity still existed within the ORC sample. To improve the resolution, we therefore tested whether different nucleotide analogs or increased concentrations of ATPγS could improve conformational uniformity (*Figure 1B*). Similar to prior observations (*Clarey et al., 2006*), 2D EM class averages of apo-ORC

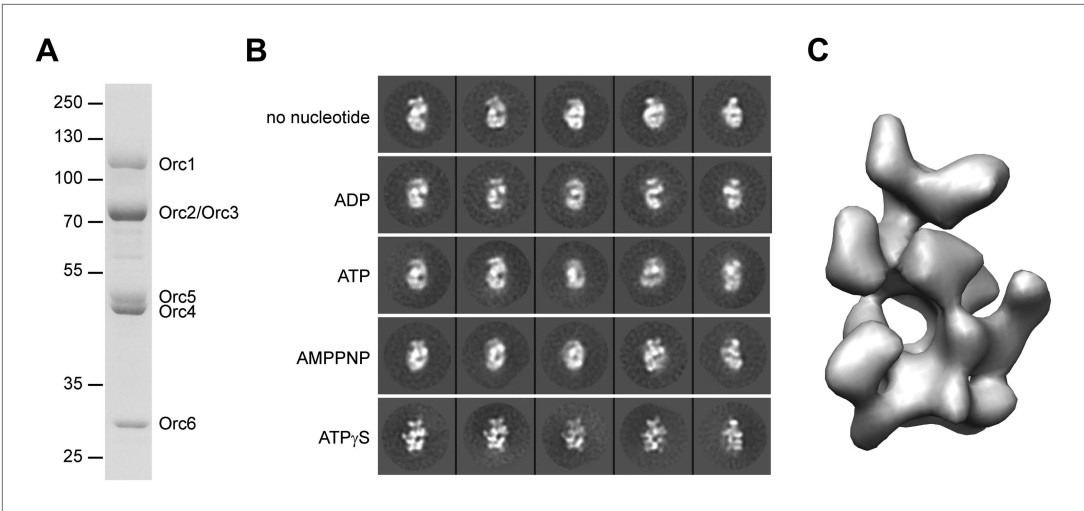

**Figure 1**. 3D structure of metazoan ORC. (**A**) Coomassie-stained SDS-PAGE gel of purified recombinant *Drosophila* ORC used for EM studies. Molecular weight markers and ORC subunits are indicated. (**B**) Nucleotides have different effects on the conformation of ORC. *Drosophila* ORC was incubated with 1 mM ADP, ATP, AMPPNP or ATPγS and analyzed by electron microscopy and compared to apo-ORC. Representative 2D class averages are shown. (**C**) Single-particle EM reconstruction of *Drosophila* ORC corrected for the absolute hand of ORC.
The following figure supplements are available for figure 1:

**Figure supplement 1**. Purification of *Drosophila* ORC.

**Figure supplement 2**. Single particle 3D EM reconstruction of *Drosophila* ORC.

**Figure supplement 3**. Determination of the absolute hand of ORC.

yielded crescent-shaped particles. Inclusion of 1 mM ATP, ADP or the non-hydrolysable ATP analog AMPPNP appeared similar to apo-ORC in 2D class averages (*Figure 1B*). However, the addition of 1 mM ATPγS revealed hitherto unobserved detailed features of ORC, indicating that ATPγS stabilizes interactions between subunits and traps ORC in a better defined conformational state. The ATPγS-dependent stabilization seen here proved more pronounced than that seen previously (*Clarey et al., 2006*), likely due to the higher concentration of nucleotide (1 mM vs 5 μM) used to ensure saturation of ORC with ATPγS. Such an effect of ATPγS on *Drosophila* ORC is somewhat similar to the nucleotide requirement observed for human ORC complex formation and stability (*Ranjan and Gossen, 2006*; *Siddiqui and Stillman, 2007*); based on this improved behavior, all subsequent EM studies of *Drosophila* ORC were performed in the presence of 1 mM ATPγS.

The more detailed features observed in 2D class averages of ATPγS-ORC suggested that it should be possible to obtain a 3D EM reconstruction of *Drosophila* ORC at a significantly improved resolution than was possible in earlier efforts. We therefore calculated a 3D EM reconstruction, using a low-pass filtered volume of the previously published *Drosophila* ORC EM structure as an initial starting model (*Figure 1—figure supplements 2A–C*). The structure was refined by iterative projection-matching to a resolution of 22 Å (as judged by the 0.5 Fourier shell correlation criterion) (*Figure 1—figure supplement 2D*), yielding forward projections of the 3D structure in excellent agreement with our reference-free class averages (*Figure 1—figure supplement 2F*). The resultant model contains contributions from a near-complete coverage of angular space (*Figure 1—figure supplement 2E*), and showed significantly more detail than the previous *Drosophila* ORC EM structure (compare *Figure 1—figure supplement 2A,C*).

Since the handedness of ORC was not previously determined experimentally, we next collected tilt-pairs of ORC and subjected them to the Freehand test (*Rosenthal and Henderson, 2003*; *Henderson et al., 2011*). This analysis showed that the initially calculated ORC structure (shown in *Figure 1—figure supplement 2C*) was of the incorrect absolute hand (*Figure 1—figure supplement 3*), leading us to flip

the hand of the EM volume for further analysis (*Figure 1C*). Taken together, our results indicate that nucleotide binding induces conformational changes within ORC that are propagated throughout the entire complex to stabilize a specific structural state.

## Subunit architecture of *Drosophila* ORC

While the higher resolution features of our new *Drospohila* EM model allowed us to delineate likely subunit borders, it still remained unknown which subunits might occupy different regions of the density. We therefore set out to localize individual subunits experimentally by EM, using purified recombinant ORC containing N-terminal MBP- and C-terminal GFP-tags on individual ORC subunits, respectively. Fusion proteins of subunits Orc1–Orc5 were able to form stable hexameric ORC complexes (*Figure 2—figure supplement 1*), allowing for the collection of electron micrographs, 2D classification, and 3D EM reconstructions. Comparison of class averages and 3D volumes of tagged ORC with those of untagged ORC revealed extra densities corresponding to the MBP- or GFP-tags, thus marking the positions of the tagged subunits (*Figure 2A*). We did not observe extra density for MBP when fused to the N-terminus of full-length Orc1 or Orc2, nor did we see GFP fused to the C-terminus of Orc2 and Orc4 (data not shown), suggesting these protein termini are flexible. However, after deletion of the N-terminal domains of Orc1 and Orc2 (which are predicted to contain disordered regions), density became visible for MBP (*Figure 2A*). During the subunit mapping process, we were unable to definitively distinguish Orc2 and Orc3 from each other, suggesting that the two subunits lie in close proximity within ORC. Collectively, our data indicate that the AAA+-like subunits Orc1–Orc5 comprise the crescent-shaped core of ORC, in the order Orc1-Orc4-Orc5-Orc2/Orc3 (when starting at the lower left of the EM structure) (*Figure 2B*).

To localize Orc6, we initially compared class averages of ORC lacking Orc6 (referred to as ORC1–5) with class averages of ORC containing Orc6 (simply referred to as ORC). Despite the reasonable molecular weight of Orc6 (29 kDa) and its stoichiometric presence in ORC purifications (*Figure 3—figure supplement 1*), 2D EM class averages of ORC and ORC1–5 looked virtually identical, suggesting that Orc6 might be flexibly attached to the rest of the complex (*Figure 3A*). To investigate the position of Orc6 further, we fused MBP or GFP to the N- or C-terminus of Orc6, respectively. In class averages of ORC containing C-terminally GFP-tagged Orc6, extra density appeared near the upper left tip of ORC, close to the position of Orc2 and Orc3 (*Figure 3A*). We did not observe density with an N-terminal MBP–Orc6 fusion, nor did we observe GFP density in 3D reconstructions (*Figure 3A* and data not shown). These data indicate that Orc6 is flexibly tethered to ORC, likely via an interaction with Orc2, Orc3, or both.

The localization pattern of Orc6 led us to hypothesize that we might be able to use it to determine the order of Orc2 and Orc3 in our EM volume by identifying Orc6's binding partner. Interactions between human and yeast Orc2, Orc3, and Orc6 have previously been investigated; however, conflicting results of subunit interactions have been reported (*Vashee et al., 2001*; *Kneissl et al., 2003*; *Chen et al., 2007*; *Matsuda et al., 2007*; *Siddiqui and Stillman, 2007*; *Sun et al., 2012*). Moreover, the interactions between the *Drosophila* proteins have not been tested. To identify the subunit that recruits *Drosophila* Orc6 into ORC, we performed pull-down assays from insect cells co-expressing pairwise combinations of *Drosophila* Orc2, Orc3, and Orc6, with one subunit containing an MBP-tag used as purification bait. As previously observed for both human and yeast proteins (*Dhar et al., 2001*; *Vashee et al., 2001*; *Ranjan and Gossen, 2006*; *Matsuda et al., 2007*), *Drosophila* Orc2 interacted with Orc3 (*Figure 3B*). This finding is consistent with our EM localization, which positioned these subunits within similar regions of ORC. Orc6, on the other hand, interacted only with Orc3 and not with Orc2 (*Figure 3B*). Since Orc6 appears to contact the subunit at the upper left end in the EM class averages, the Orc3–Orc6 interaction allowed us to refine the subunit order of the core AAA+ subunits to Orc1-Orc4-Orc5-Orc2-Orc3 for *Drosophila* ORC (*Figure 2B*).

## The C-terminus of Orc6 mediates its recruitment into ORC

Metazoan Orc6 is composed of three domains: an N-terminal and middle domain, both of which have homology to the cyclin-box folds of transcription initiation factor IIB (referred to as TFIIB domain A and B), and a C-terminal region of unknown structure (*Figure 4A*, *Figure 4—figure supplement 1*) (*Chesnokov et al., 2003*; *Liu et al., 2011*). The C-terminal domain of metazoan Orc6 has been thought to be mainly important for non-replicative functions of Orc6 in cytokinesis and daughter cell abscission, and in the case of *Drosophila* Orc6, the C-terminal domain also has been

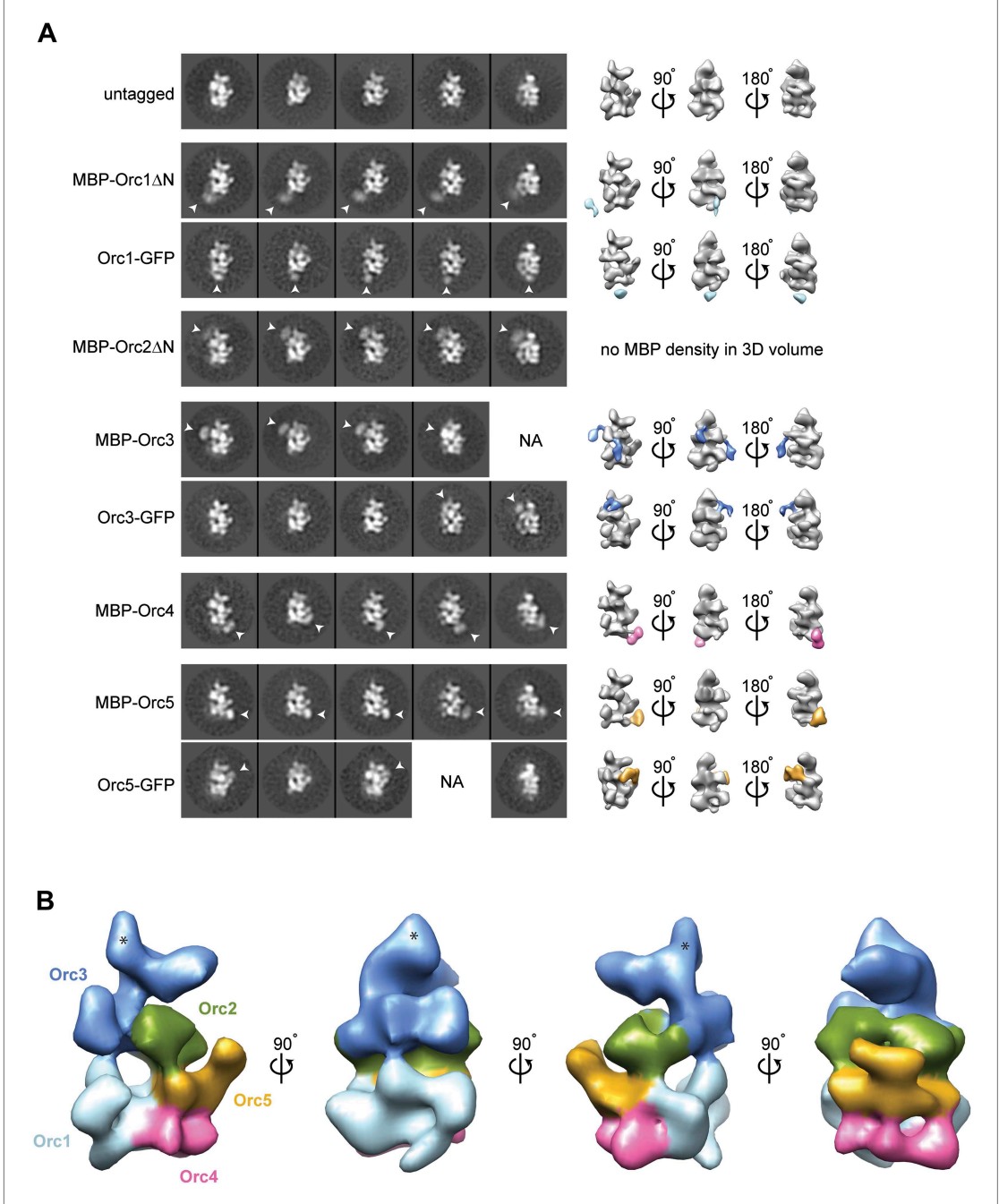

**Figure 2**. Arrangement of subunits within metazoan ORC. (**A**) Localization of AAA+ subunits Orc1–Orc5 within ORC by EM. 2D class averages of ORC containing N-terminally MBP-tagged or C-terminally GFP-tagged Orc1–5 subunits are compared to representative averages of untagged ORC. Additional densities observed that can be attributed to MBP- or GFP-tags are noted by arrowheads and mark the position of the respective subunit within ORC. 3D EM reconstructions of MBP- or GFP-tagged ORCs further pinpoint the localization of Orc1–Orc5 within ORC. With the exception of MBP–Orc2ΔN, MBP- or GFP-densities (highlighted in color) were also observed in 3D EM reconstructions of MBP- or GFP-tagged ORCs. No additional density was observed in 2D class averages for ORC containing N-terminally MBP-tagged full-length Orc1 and Orc2 nor for C-terminally GFP-tagged Orc2 and Orc4, indicating conformational flexibility of the tag (data not shown). NA—this view was not observed in 2D class averages. (**B**) Surface coloring of the *Dm*ORC EM volume based on subunit localization. The asterisks marks density that likely contains a domain inserted between the AAA+-like domain and the winged helix domain in Orc3 (*Figure 5*).

The following figure supplements are available for figure 2:

**Figure supplement 1**. Purification of ORC with tagged Orc1–5 subunits for EM localization.

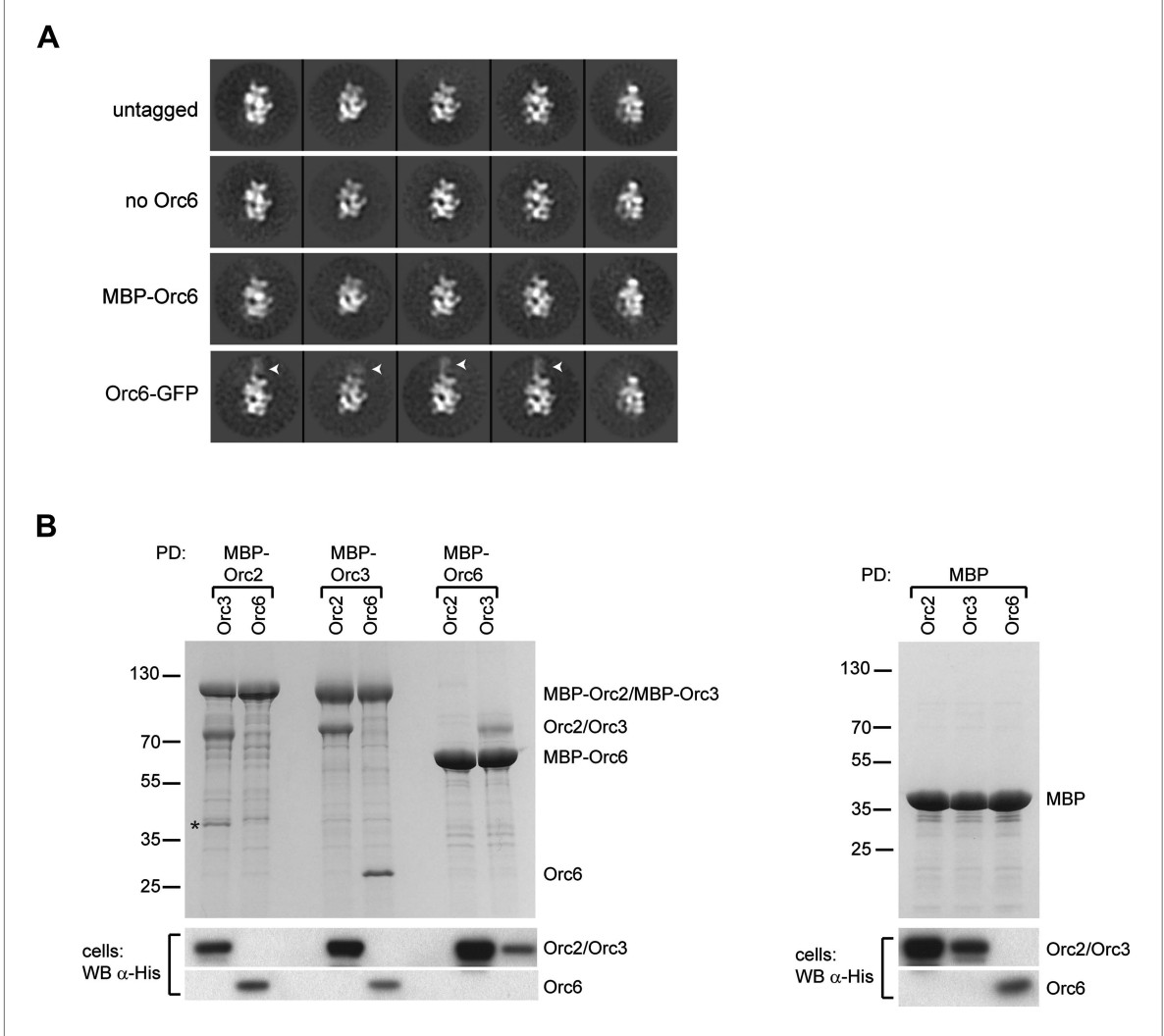

**Figure 3**. Orc6 localizes near Orc2 and Orc3 within *Drosophila* ORC but is conformationally flexible. (**A**) ORC containing N-terminally MBP-tagged Orc6 or C-terminally GFP-tagged Orc6 was purified and subjected to EM analysis. Representative 2D class averages are shown. *Drosophila* ORC with untagged Orc6 is shown for comparison. Additional density corresponding to the tag (arrowheads) is only observed for the Orc6-GFP and not for MBP–Orc6. Class averages of ORC lacking Orc6 subunit reveal no difference density compared to ORC containing Orc6, indicating that Orc6 is flexibly tethered to ORC. (**B**) *Drosophila* Orc6 interacts with Orc3 but not with Orc2. MBP–Orc2, MBP–Orc3 and MBP–Orc6 fusions were co-expressed in High5 cells with 6 × His-tagged Orc2, Orc3 and Orc6 and subjected to pull-downs (PD) using amylose resin. Orc3 interacts with both Orc2 and Orc6, whereas no interaction is observed between Orc2 and Orc6 as assessed by SDS-PAGE and Coomassie staining. In addition, no interaction is observed between MBP alone and any of the ORC subunits. The asterisk denotes a degradation product of MBP–Orc2. Western blot (WB) analysis of whole cell extracts for 6 × His tagged subunits (lower panels) demonstrates that they are expressed as expected.

The following figure supplements are available for figure 3:

**Figure supplement 1**. Purification of ORC complexes for EM localization of the Orc6 subunit.

shown to interact with the septin Pnut (*Prasanth et al., 2002*; *Chesnokov et al., 2003*; *Bernal and Venkitaraman, 2011*). Surprisingly, however, we found that the C-terminal domain of Orc6 proved to be both essential and sufficient for Orc3 binding when co-expressed in insect cells (*Figure 4B*, *Figure 4—figure supplement 2A* for controls). The necessity of the C-terminal domain was not limited to binary protein interactions, but was also observed either when all ORC subunits were co-expressed in insect cells (*Figure 4—figure supplement 2B*), or when purified full-length or C-terminal truncated Orc6 proteins were mixed with ORC1–5 in vitro and subjected to pull-down assays (*Figure 4C*).

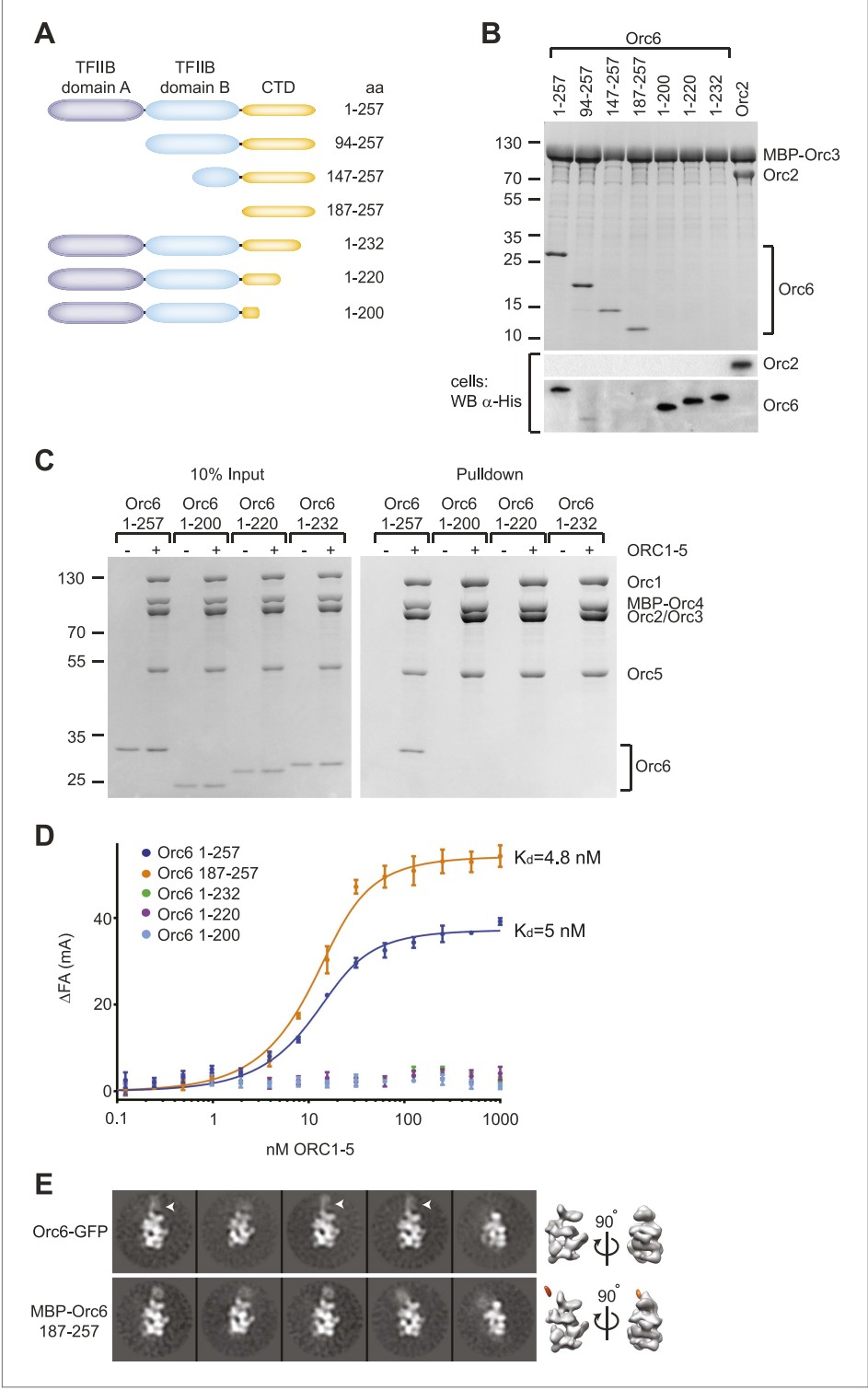

**Figure 4**. *Drosophila* Orc6 is recruited into ORC via interaction of its C-terminal domain with Orc3. (**A**) Schematic of domain organization of Orc6 and of the N- and C-terminal deletion constructs of Orc6 used in this study. (**B**) The C-terminal domain (CTD) of Orc6 is essential and sufficient for the interaction of Orc6 with Orc3. His-tagged full-length Orc6 (1–257) and Orc6 deletion constructs were co-expressed with MBP-tagged Orc3 in High5 cells and analyzed for interactions by pull-downs using amylose resin. Bound proteins were analyzed by SDS-PAGE and Coomassie-staining. Western blot analysis of whole cell extracts (lower panel) shows that C-terminally truncated Orc6 proteins are expressed at similar levels to full-length wild type Orc6. Despite the clear pull-down results for the N-terminally truncated Orc6

*Figure 4. Continued on next page*

*Figure 4. Continued*

proteins, we were not able to detect two of the proteins (Orc6 147–257 and 187–257) in whole cell extracts, possibly due to expression levels too low to be detected with our Western blot conditions. (**C** and **D**) The C-terminal domain of Orc6 is essential and sufficient for recruitment of Orc6 into ORC. Purified ORC1–5 (lacking Orc6 subunit) was incubated in vitro with purified full-length Orc6, Orc6 lacking C-terminal residues (Orc6 1–232, Orc6 1–220 and Orc6 1–200), and the isolated Orc6 C-terminal domain (Orc6 187–257). Formation of ORC containing all six subunits was tested in pull-downs using (**C**) MBP-tagged Orc4 as bait and by (**D**) fluorescence anisotropy after N-terminally labeling Orc6 proteins with Alexa Fluor-488. C-terminal deletions of Orc6 abolish recruitment of Orc6 into ORC in vitro, whereas the C-terminal domain of Orc6 binds to ORC1–5 with nanomolar affinity. Binding data of full-length Orc6 and the Orc6 C-terminal domain (Orc6 187–257) were fit to the quadratic equation describing single-site binding under ligand depletion. (**E**) The localization of the C-terminal domain of Orc6 is indistinguishable from the localization of full-length Orc6 within ORC. Purified ORC containing full-length Orc6-GFP or MBP–Orc6 187–257 was analyzed by negative stain EM. 2D image analysis revealed extra densities (arrow heads) in similar positions that correspond to GFP and MBP tags on the respective Orc6 constructs. Density corresponding to the affinity tag was observed in the 3D volume of ORC containing MBP–Orc6 187–257, further pinpointing the localization of Orc6 within ORC.

The following figure supplements are available for figure 4:

**Figure supplement 1**. The Orc6 TFIIB-like domains as well as residues in the C-terminal domain containing the Meier–Gorlin syndrome mutation in human Orc6 are conserved across eukaryotes, including fungi.

**Figure supplement 2**. C-terminal truncations of *Drosophila* Orc6 do not associate with ORC1–5 when all subunits are co-expressed in insect cells.

To rule out the possibility that our pull-downs might miss a potential weak interaction between ORC1–5 and Orc6 lacking the C-terminal domain, we performed equilibrium binding experiments using fluorescence anisotropy and fluorescently-labeled Orc6. Full-length Orc6 bound to ORC1–5 with apparent low-nanomolar affinity ($K_{d, app}$ = 5.0 ± 0.7 nM) (***Figure 4D***). Consistent with the pull-down results, C-terminally truncated Orc6 proteins did not bind ORC1–5 up to concentrations of 1 μM ORC1–5 (***Figure 4D***). In contrast, the C-terminal 71 amino acid residues of Orc6 bound to ORC1–5 with an affinity similar to that of full-length Orc6 ($K_{d, app}$ = 4.8 ± 0.6 nM). In addition, the Orc6 C-terminal region co-purified with ORC1–5 during several chromatography steps (***Figure 3—figure supplement 1***, lane 5), and when examined by EM, an MBP-fusion of the C-terminal domain of Orc6 localized to the same region as full-length Orc6 in 2D class averages of ORC, indicating that Orc6 did not dissociate from ORC at concentrations used for EM (30 nM ORC) (***Figure 4E***). Collectively, these results demonstrate that Orc6 is recruited into ORC via interaction of its C-terminal domain with Orc3, and not through its TFIIB-like domains.

We next asked which region of Orc3 interacts with Orc6. In comparing Orc3 protein sequences from various eukaryotes with archaeal Orc1/Cdc6 proteins, we noticed that Orc3 contains an additional domain inserted between its AAA+-like and winged-helix domains that is not found in archaeal homologs (***Figure 5A,B***, ***Figure 5—figure supplement 1***). This insertion is conserved across eukaryotic Orc3 orthologs, varies in length from 70 amino acids in *S. cerevisiae* to 190 amino acids in *Drosophila,* and is not found in Orc1, Orc2, Orc4, or Orc5 (***Figure 5—figure supplement 1***). Since this extra domain is specific to Orc3, we hypothesized that it might interact with Orc6, specifically with the Orc6 C-terminus. To test this premise, we co-expressed the Orc3 insertion as an MBP fusion protein with full-length or truncated Orc6 and performed pull-down experiments. The Orc3 insertion was able to interact with both full-length Orc6 and all N-terminal Orc6 truncations (***Figure 5C***). Conversely, truncating the C-terminus of Orc6 abolished the interaction with the Orc3 insertion. These results mirror our observations with both full-length Orc3 and with ORC1–5, and further demonstrate that the C-terminus of Orc6 helps to tether Orc6 to ORC. Moreover, whereas full-length Orc3 was able to interact with Orc2 in addition to Orc6, the Orc3 insertion was unable to do so, suggesting the Orc2–Orc3 interaction is mediated through the two subunits' AAA+-like and/or winged-helix domains (***Figure 5D***).

## Conserved amino acids in the C-terminus of Orc6, including the tyrosine mutated in MGS patients, mediate Orc6 recruitment into ORC and are required for MCM2–7 loading onto chromatin

Although the overall function of Orc6 has remained somewhat debatable, the subunit is one of the pre-RC components recently shown to be mutated in patients with Meier-Gorlin syndrome (MGS)

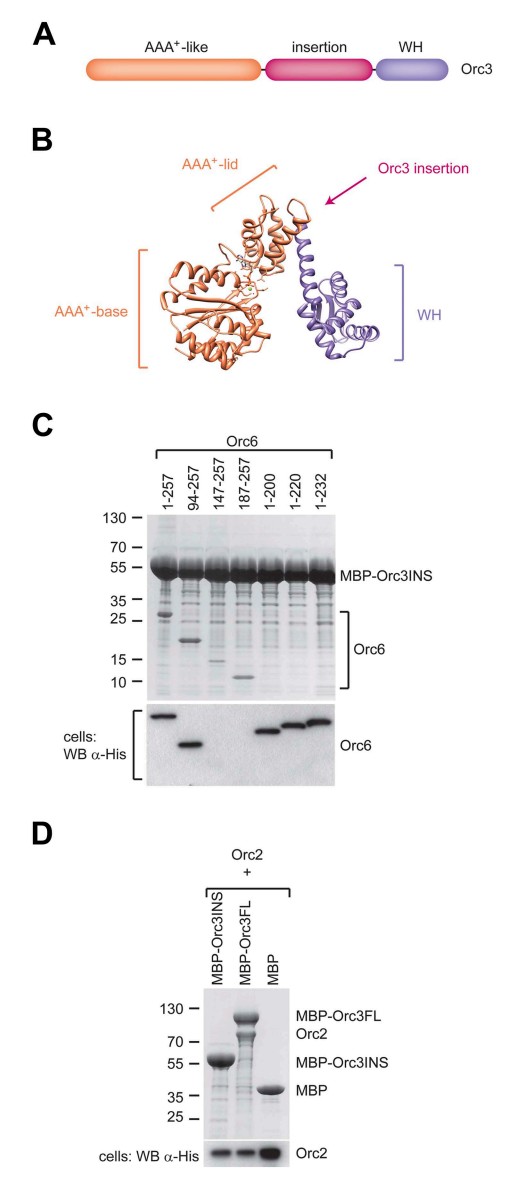

**Figure 5**. A conserved domain inserted between the AAA⁺-like domain and the winged helix (WH) domain of Orc3 binds the C-terminal domain of Orc6 and recruits Orc6 into ORC. (**A**) Schematic domain organization of Orc3. (**B**) The position of the Orc3 insertion maps between the AAA⁺-like domain and the winged helix domain in the crystal structure of archaeal Orc1–1 (PDB code 2qby chain A) (*Dueber et al., 2007*). (**C**) The Orc3 insertion is sufficient for Orc6 binding. *Drosophila* Orc3 containing residues 371–559 (Orc3INS) was co-expressed as an N-terminal MBP-fusion protein with His-tagged full-length and truncated versions of *Drosophila* Orc6 in High5 cells and their interaction probed by pull-downs using amylose resin. (**D**) The insertion in Orc3 does not interact with Orc2. N-terminal MBP-fusion proteins of *Drosophila* full-length Orc3 (Orc3FL), the Orc3 insertion (Orc3INS) or MBP alone were co-expressed with

*Figure 5. Continued on next page*

(*Bicknell et al., 2011a*). Intriguingly, the MGS mutation in human Orc6 is a missense alteration that maps to the protein's C-terminal domain, changing a tyrosine at position 232 to serine. The tyrosine is located in a highly conserved, predicted helical element, and is itself highly conserved among eukaryotic Orc6 orthologs, including budding yeast (*Figure 6A*, *Figure 4—figure supplement 1*). Since our findings show that the C-terminal domain of Orc6 is essential for the recruitment of Orc6 into *Drosophila* ORC, we hypothesized that the MGS mutation may interfere with this function. To test this assumption, we introduced the corresponding Y225S mutation into *Drosophila* Orc6 and asked whether it could affect either the binary Orc3–Orc6 interaction or hexameric ORC formation. In addition, we included another mutant, W228A/K229A, which substitutes an invariant tryptophan close to Y225 and a highly conserved basic amino acid for alanine. Orc6-W228A/K229A is not able to rescue an Orc6 deletion in *Drosophila* to support viability in vivo, and cells bearing the two mutations showed reduced BrdU incorporation in brain neuroblasts, suggesting defects in DNA synthesis (*Balasov et al., 2009*). When co-expressed in insect cells with MBP–Orc3, drastically reduced levels of Orc6-Y225S and no Orc6-W228A/K229A were pulled-down (*Figure 6B*). Similar results were obtained when the Orc3 insertion was used instead of full-length Orc3 (*Figure 6—figure supplement 1A*). Interestingly, the Orc3–Orc6 interaction was not stabilized by additional co-expression of Orc2, suggesting that Orc2 binding to Orc3 has no effect on the Orc3–Orc6 interaction in *Drosophila* (*Figure 6—figure supplement 1B*).

To address whether the MGS-like mutation and the WK mutant affected hexameric ORC formation, we next tested whether purified wild-type or mutant *Drosophila* Orc6 could bind to purified ORC1–5 in vitro. Neither Orc6-Y225S nor Orc6-W228A/K229A bound to ORC1–5 in pull-down experiments (*Figure 6C*). Orc6-Y225S incorporation into ORC was also reduced when it was co-expressed with Orc1–Orc5 subunits in insect cells (*Figure 4—figure supplement 2B*, lane 8). In equilibrium-binding experiments using fluorescence anisotropy, Orc6-W228A/K229A showed no binding to ORC1–5 (*Figure 6D*). Orc6-Y225S was able to bind weakly to Orc1–5; however, the affinity was reduced at least 100-fold compared to wild type Orc6, and similar results were obtained for the MGS-like mutant when only the C-terminal domain of Orc6 was used in binding experiments (*Figure 6—figure supplement 1C*). Together, these results show that highly conserved amino acid

*Figure 5. Continued*

6 × His-tagged *Drosophila* Orc2 in High5 cells and subjected to amylose pull-downs. Coomassie-stained SDS-PAGE gels are shown in (**C** and **D**). Western blot analysis of whole-cell extract using an anti-His antibody (lower panels in **C** and **D**) demonstrates that lack of Orc2 or Orc6 pull-down is not due to lack of their expression. As in *Figure 4B*, we were not able to detect Orc6 147-257 and Orc6 187-257 in cell extracts despite being clearly visible in pull-downs, likely because of low expression levels.

The following figure supplements are available for figure 5:

**Figure supplement 1**. Eukaryotic Orc3 contains a domain inserted between its AAA+-like and winged helix domains that is not present in archaeal Orc1/Cdc6, nor in eukaryotic Orc1, Orc2, Orc4 or Orc5.

residues in the C-terminal domain of *Drosophila* Orc6, including the tyrosine mutated in human Orc6 in Meier-Gorlin syndrome, mediate the recruitment of Orc6 into ORC.

In flies, the Orc6 C-terminus is essential for viability, as GFP–Orc6 transgenes containing mutations in the C-terminus are (in contrast to wild type GFP–Orc6) unable to rescue an *orc6*-null allele (*Balasov et al., 2009*). Neuroblasts of such Orc6 mutant flies incorporate BrdU less efficiently than cells expressing wild-type Orc6, suggesting that they suffer from defects in DNA replication (*Balasov et al., 2009*). We re-investigated flies carrying these transgenes to test whether the previously observed phenotypes are due to defects in hexameric ORC formation in vivo, and to determine the physiologic consequences of our in vitro results. We expressed GFP–Orc6 wild type, GFP–Orc6 D224A/Y225A or GFP–Orc6 W228A/K229A transgenes in flies and asked whether they associate with other ORC subunits in vivo. ORC was immunoprecipitated with an anti-Orc2 antibody from ovary extract and, as expected, immunoprecipitates were found to contain Orc3, Orc5, and GFP–Orc6 wild type (*Figure 7A*). In contrast, the amounts of GFP–Orc6 mutants that were immunoprecipitated were greatly reduced compared to wild type GFP–Orc6, both in Western blots and when GFP absorbance was measured directly (*Figure 7A,B*). Thus, mutations in the conserved C-terminus of Orc6 result in a reduction of ORC hexamer formed both in vitro and in vivo.

The reduced levels of hexameric ORC may be expected to lead to impaired pre-RC formation. To test this premise, we analyzed chromatin association of a member of the MCM2–7 complex, Mcm4, in *Drosophila* larval brains expressing either wild-type or mutant GFP–Orc6 transgenes. After salt extraction of larval brain extracts, the amount of Mcm4 in the pellet fraction was greatly reduced when mutant transgenes were expressed as the sole source of Orc6 (*orc6^35^/orc6^35^*) compared to larval brains expressing the wild-type GFP–Orc6 transgenes (*Figure 7C*). As expected, heterozygous larval brains, which express wild-type, endogenous Orc6, did not show a Mcm4 chromatin recruitment defect. Thus, conserved C-terminal residues of Orc6 are not only required for hexameric ORC formation but also for MCM loading onto chromatin, explaining the reduced DNA synthesis previously observed in mutant flies (*Balasov et al., 2009*).

Although Orc6, including the C-terminal helix, is well conserved among metazoans, human Orc6 does not appear to tightly associate with human ORC1–5 (*Dhar and Dutta, 2000*; *Dhar et al., 2001*; *Vashee et al., 2001*; *Giordano-Coltart et al., 2005*; *Ranjan and Gossen, 2006*; *Siddiqui and Stillman, 2007*). We therefore set out to test whether the C-terminus of human Orc6 is also important for assembly of the subunit with the rest of human ORC and whether the Orc6 MGS mutation might affect this interaction. We purified recombinant human ORC1–5 from insect cells, incubated it with purified human Orc6 proteins in vitro, and assayed ORC hexamer formation by pull-downs targeting MBP-tagged Orc4 within ORC1–5. Although weaker than seen for *Drosophila*, we nonetheless clearly observed an interaction between wild-type Orc6 and human ORC1–5, particularly when an excess of Orc6 was used (*Figure 8A*). In contrast, human Orc6 containing the MGS mutation (Orc6-Y232S) did not bind appreciably to ORC1–5 (*Figure 8A*). Similar results were obtained when human ORC subunits were co-expressed in insect cells (*Figure 8B*). As seen here with the *Drosophila* protein, human Orc6 has been reported to interact with Orc3 (*Figure 8C*) (*Vashee et al., 2001*; *Siddiqui and Stillman, 2007*). Importantly, human Orc6 containing the MGS mutation could not bind Orc3 in either the absence or presence of Orc2 (*Figure 8D*). These data indicate that the Orc6 C-terminus is functionally conserved in metazoans, and that the amino acid mutated in MGS patients is critical for formation of a stable hexameric initiator complex.

## Functional conservation of the Orc6 C-terminus between metazoans and yeast

Metazoan and *Schizosaccharomyces pombe* Orc6 have been reported to share little sequence homology with *S. cerevisiae* Orc6, and unique functions have been reported for metazoan and budding

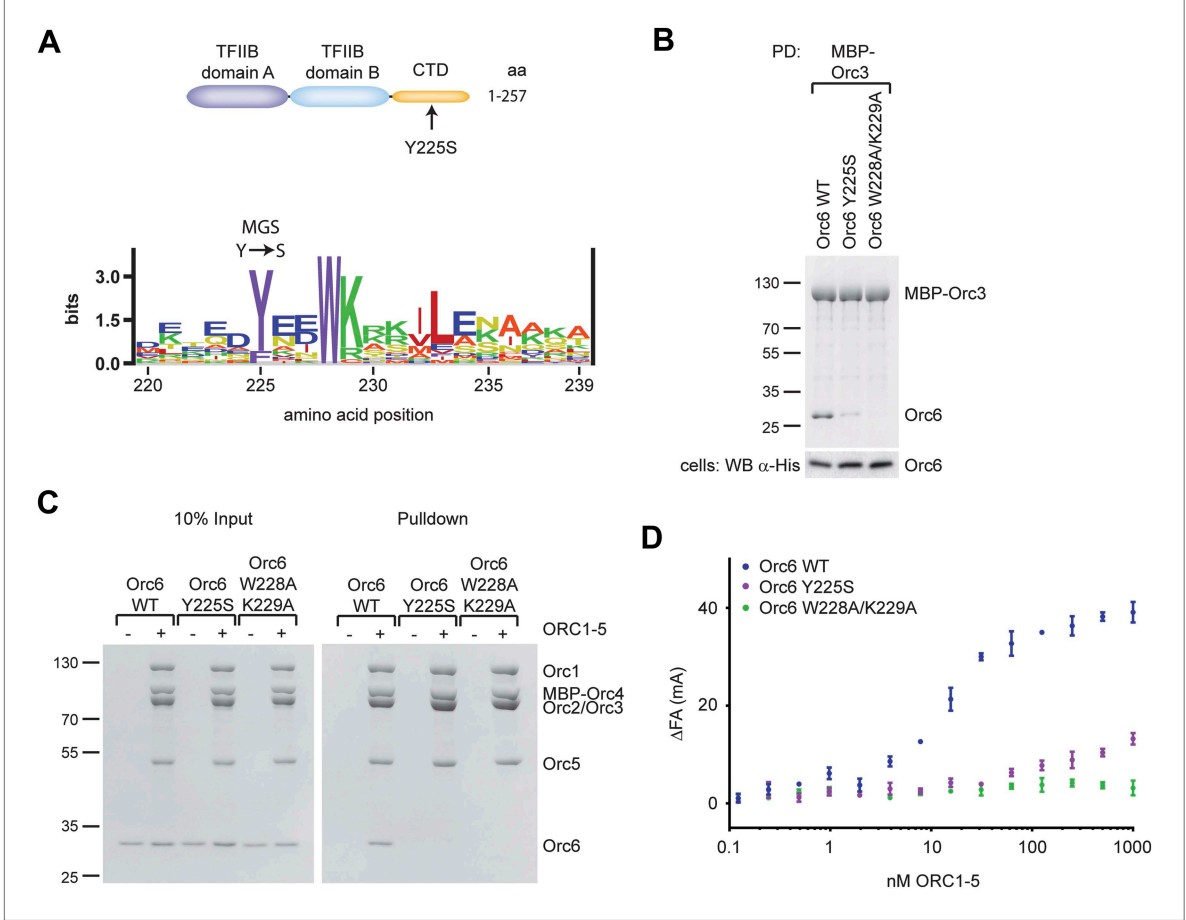

**Figure 6**. A *Drosophila* Orc6 mutation corresponding to the Meier-Gorlin-Syndrome mutation in human Orc6 weakens the interaction of Orc6 with Orc3 and diminishes its recruitment into ORC. (**A**) The tyrosine mutated in Meier-Gorlin syndrome (MGS) is a well conserved residue and maps to the C-terminal domain of Orc6. A sequence LOGO (***Workman et al., 2005***) of the Orc6 C-terminus is shown. The amino acid residue corresponding to the human MGS mutation was mutated in *Drosophila* Orc6 (Y225S). (**B**) Mutations in the C-terminus of *Drosophila* Orc6, including the MGS mutation, reduce the affinity of Orc6 for Orc3. Wild-type (WT) Orc6, the Y225S mutant or the W228A/K229A double-mutant Orc6 were co-expressed with MBP–Orc3 in High5 cells and tested for interaction with Orc3 by pull-down analysis using amylose beads and MBP–Orc3 as bait. Coomassie-stained SDS-PAGE gels are shown for pull-downs and a Western blot (lower panel) is shown to confirm Orc6 expression in insect cells. (**C** and **D**) C-terminal mutations in *Drosophila* Orc6 interfere with recruitment of Orc6 into ORC in vitro. Wild-type Orc6, Orc6-Y225S and Orc6-W228A/K229A were purified and tested for association with ORC1–5 into heterohexameric ORC using pull-downs targeting MBP–Orc4 (**C**) or by fluorescence anisotropy using N-terminally Alexa Fluor-488 labeled Orc6 (**D**).

The following figure supplements are available for figure 6:

**Figure supplement 1**. Mutations in the C-terminal domain of *Drosophila* Orc6, including the MGS-like mutation, abrogate binding of Orc6 to the Orc2–Orc3 subcomplex, to the Orc3 insertion (Orc3INS) and to ORC1–5.

yeast Orc6 (***Chesnokov et al., 1999***; ***Moon et al., 1999***; ***Dhar and Dutta, 2000***; ***Duncker et al., 2009***). For example, metazoan but not budding yeast Orc6 binds DNA, while yeast Orc6 has been reported to bind Cdt1 (***Balasov et al., 2007***; ***Chen et al., 2007***; ***Liu et al., 2011***). This lack of similarity is surprising considering the generally high conservation of other licensing factors and the central role Orc6 is reported to play during replicative helicase loading in *S. cerevisiae* (***Takara and Bell, 2011***; ***Fernandez-Cid et al., 2013***; ***Frigola et al., 2013***). However, in the course of analyzing our sequence alignments of Orc6, we noticed that significant homology actually exists between metazoan, *S. pombe*, and *S. cerevisiae* Orc6 (***Figure 4—figure supplement 1***). This homology includes both the TFIIB repeat domains and the conserved helix in the Orc6 C-terminal domain that harbors the MGS mutation, indicating that all of the functional domains present in metazoan Orc6 are conserved in budding yeast

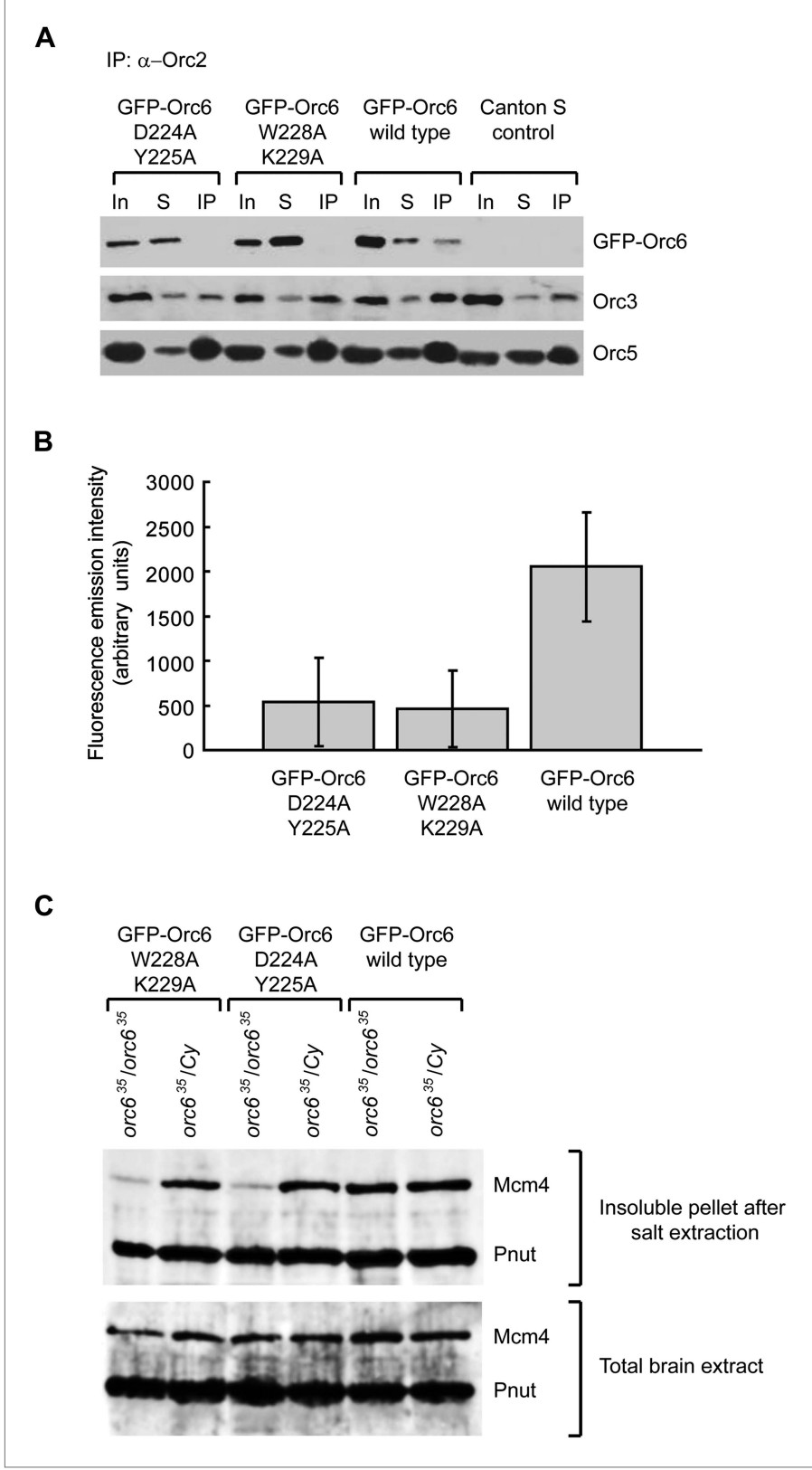

**Figure 7**. Mutations of conserved amino acids in the C-terminal domain of *Drosophila* Orc6 result in a loss of its association with ORC and in reduced MCM chromatin association in vivo in flies. (**A**) Western immunoblotting analysis of ORC complexes precipitated using anti-Orc2 antibody from extracts isolated from fly ovaries expressing

*Figure 7. Continued on next page*

*Figure 7. Continued*

different GFP-tagged Orc6 mutant proteins: GFP–Orc6 D224A/Y225A, GFP–Orc6 W228A/K229A and GFP–Orc6 WT (wild type). GFP–Orc6 fusion proteins were detected with anti-GFP monoclonal antibody, while Orc3 and Orc5 subunits were detected using anti-Orc3 or anti-Orc5 antibodies. Ovary extract (In), supernatant after immunoprecipitations (S) and immunoprecipitated material (IP) are shown for each transgene. (**B**) Quantitation of immunoprecipitated GFP from ovary extracts containing different Orc6 mutants. The Y-axis shows fluorescence emission intensity. Mean and standard deviation from three independent experiments are plotted (p≤0.05). (**C**) Chromatin association of Mcm4, a member of the MCM2–7 complex, is reduced in *orc6*-null fly larvae expressing Orc6 mutants. Brains of homozygous, orc6-null (*orc6³⁵/orc6³⁵*) or heterozygous (*orc6³⁵/Cy*) *Drosophila* larvae expressing GFP–Orc6 transgenes were isolated and subjected to salt extraction to solubilize cellular proteins. Insoluble and chromatin associated proteins were pelleted by centrifugation and analyzed by Western blotting using Mcm4 polyclonal antibodies. Pnut was used as a loading control. Total brain extracts were also analyzed to ensure equal Mcm4 expression in larvae expressing Orc6 transgenes.

Orc6 (*Figure 4—figure supplement 1*). However, in contrast to *S. pombe* and metazoan Orc6, budding yeast Orc6—as well as related *Saccharomycotina* (*Candida albicans*) and *Pezizomycotina* (*Aspergillus fumigatus* and *Neurospora crassa*)—contain an insertion of ~100–200 amino acids between the two TFIIB-like helical domains, an addition that may have complicated former analyses (*Figure 4—figure supplement 1*). This insertion contains phosphorylation sites for cyclin-dependent kinases, as well as an RXL motif reported to bind the cyclin Clb5, and has been implicated in contributing to the prevention of DNA re-replication (*Nguyen et al., 2001*; *Wilmes et al., 2004*; *Chen and Bell, 2011*).

Since the C-terminal domain of metazoan Orc6 associates with ORC by binding to Orc3, we reasoned that the conservation of this region in budding yeast might extend to this interaction as well. Despite previous studies, the recruitment mechanism of *S. cerevisiae* Orc6 into ORC has been controversial. The C-terminal 62 amino acids of *S. cerevisiae* Orc6 have been shown to bind ORC1–5, likely through interaction with Orc3 or Orc5 (*Chen et al., 2007*). However, Orc6 has also been shown to bind Orc2 but not Orc3 or Orc5 (*Sun et al., 2012*). Hence, we investigated whether *S. cerevisiae* Orc6 could bind to Orc2 or Orc3, and whether a mutation of the conserved tyrosine, which corresponds to the MGS mutations in human Orc6, might affect this interaction. In pull-down experiments from insect cell lysate co-expressing pairwise combinations of budding yeast Orc2, Orc3, and Orc6, we observed a clear interaction between Orc2 and Orc3, and between Orc3 and Orc6 (*Figure 9A*). Orc2, when expressed without Orc3, was typically degraded, preventing an assessment of whether it efficiently bound Orc6 or not. We next tested whether a mutation of the conserved MGS tyrosine to serine (Y418S) could support the binding of *S. cerevisiae* Orc6 to Orc3 (the MGS tyrosine in *S. cerevisiae* Orc6 highlighted by our alignment differs from that proposed previously [Tyr277], due to a misalignment of the region in the original study [*Bicknell et al., 2011a*]). Strikingly, the MGS-like mutation in *S. cerevisiae* Orc6 abolished the interaction with Orc3 (*Figure 9B*). However, in contrast to what we observed with *Drosophila* and human proteins, the MGS-like mutation was not sufficient to prevent Orc6 binding to the Orc2–Orc3 subcomplex or to ORC1–5 (*Figure 9C,D*). Because Orc6 has been previously reported to interact with Orc2 (*Sun et al., 2012*), our result suggests that *S. cerevisiae* Orc6 is cooperatively recruited into ORC through interaction with both Orc2 and Orc3. In agreement with this interpretation, Orc6 was still recruited into ORC lacking the Orc2 subunit, albeit at reduced levels, whereas Orc6 recruitment was completely lost in the absence of Orc3 (omission of Orc3 further led to concomitant loss of Orc2) (*Figure 9E*). Thus, unlike in metazoans, *S. cerevisiae* ORC appears to possess two Orc6 binding sites, one in Orc2 and one in Orc3. Although the interaction with Orc3 is mediated by the C-terminal domain of Orc6, it is not clear which domain is responsible for binding to Orc2. However, since the Orc2–Orc6 interaction appears unique to *S. cerevisiae* (compare *Figures 3B, 8 and 9*), we propose this contact may be mediated by a yeast-specific domain inserted between the two TFIIB-like domains of Orc6, which may in turn bind to the non-conserved N-terminal domain of yeast Orc2 that has been implicated previously in Orc6 interactions (*Sun et al., 2012*). Consistent with our interpretation, an MGS-like mutation in yeast Orc6 supports yeast growth, whereas premature stop codons that result in C-terminally truncated Orc6 proteins do not (*Figure 9—figure supplement 1*). Taken together, our results reconcile previously conflicting data regarding Orc6 recruitment into ORC and highlight the importance of the conserved C-terminal segment of Orc6 in the formation of the hexameric ORC assembly through Orc6 binding to Orc3. This interaction appears functionally relevant,

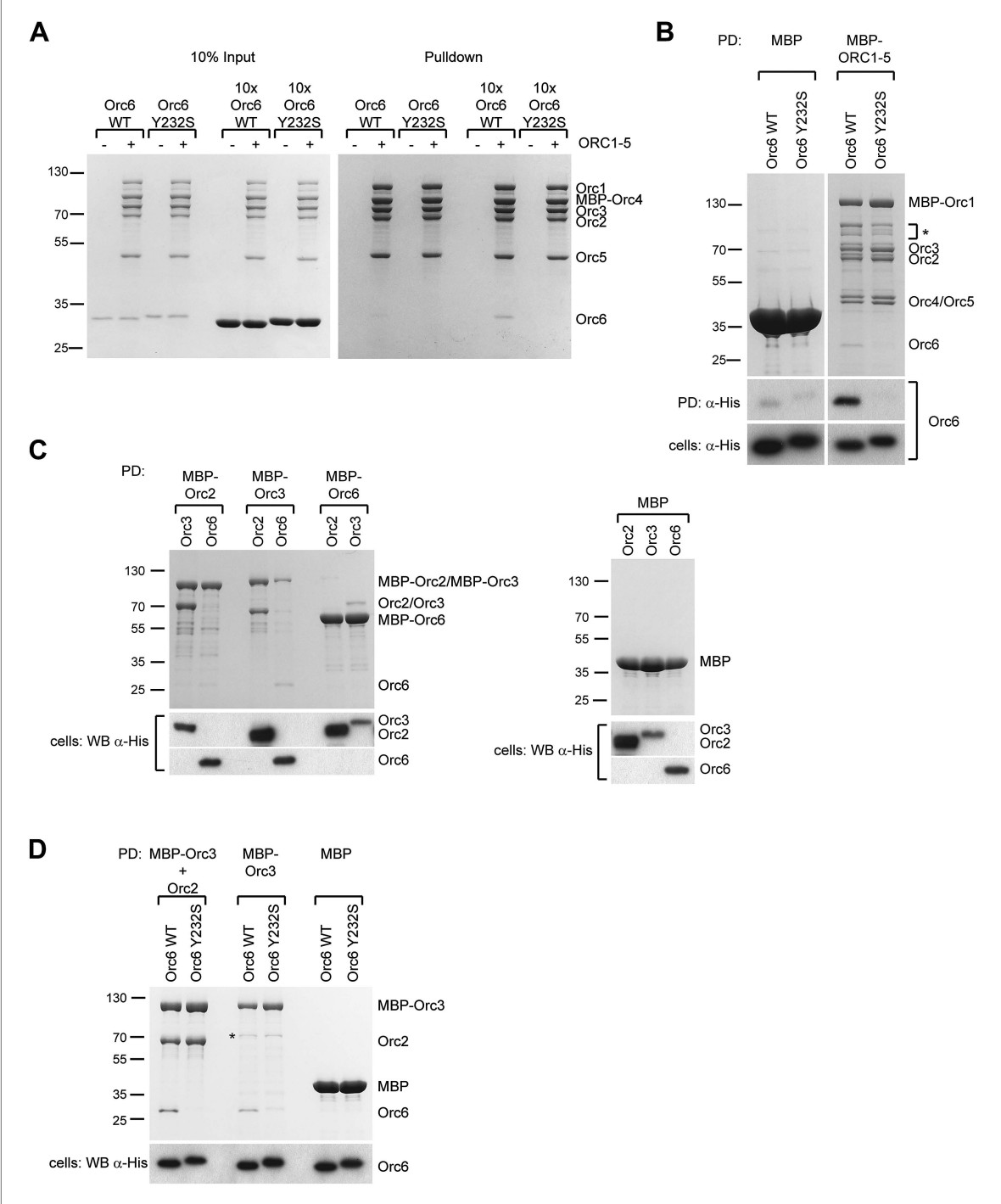

**Figure 8**. The Orc6–Orc3 interaction is conserved in humans and is affected by a mutation in human Orc6 that is found in MGS patients. (**A**) Human wild-type (WT), but not MGS mutant (Y232S), Orc6 binds to human ORC1–5 in vitro. Purified Orc6 was incubated with recombinant ORC1–5 at either equimolar ratios or with a 10 × excess of Orc6. Orc6 binding to ORC1–5 was analyzed by pull-downs using amylose beads that bind MBP–Orc4. Contrary to wild-type Orc6, MGS-mutant Orc6 does not associate with ORC1–5, even when a 10-fold excess of Orc6 is used in binding reactions. (**B**) Human Orc6 containing the MGS mutation does not co-purify with human ORC1–5 when co-expressed in insect cells. High5 cells expressing human ORC1–5 and either Orc6-WT or Orc6-Y232S were used in pull-downs targeting MPB-Orc1. As a negative control, Orc6-WT or Orc6-Y232S were co-expressed with MBP. Only Orc6-WT is recovered from beads, and only when Orc1–Orc5 are co-expressed. The presence of Orc6 was detected by Western blotting in both pull-downs (PD) and whole-cell extracts (lower panel). The asterisk marks degradation products of MBP–Orc1. (**C**) Subunit interactions within the human Orc2-Orc3–Orc6 subcomplex. MBP-tagged human ORC subunits were co-expressed in High5 cells with 6 × His-tagged subunits in a pairwise

*Figure 8. Continued on next page*

*Figure 8. Continued*

manner and probed for interaction by amylose pull-downs. Interactions are observed between human Orc3 and Orc2 as well as between Orc3 and Orc6. As a control, 6 × His-tagged subunits were co-expressed with MBP alone, and no interaction is observed with MBP. Western blots (lower panels) verified that all His-tagged subunits are present in whole cell extracts. (**D**) The MGS mutation in human Orc6 decreases the affinity of human Orc6 for human Orc3 and reduces Orc6 recruitment into the Orc2-Orc3-Orc6 subcomplex. Human ORC subunits were co-expressed in High5 cells as indicated and analyzed by amylose pull-downs for binding. The asterisk marks a degradation product of MBP–Orc3. Western blot of cell lysate shows that wild-type and mutant Orc6 are expressed in insect cells at similar levels (lower panel). Coomassie-stained gels are shown for pull-downs in (**A–D**). Orc6-Y232S always migrated slightly slower than wild-type Orc6 in SDS-PAGE gels. Mass spectrometry confirmed that the molecular weight of wild type and Y232 Orc6 are as expected based on their amino acid sequence.

as Orc6 C-terminal amino acid residues are required for yeast viablility (*Figure 9—figure supplement 1*) (*Chen and Bell, 2011*), and yeast Cdt1 fused to the C-terminal domain of Orc6 alone is able to load MCM2–7 to replication origins in vitro (*Chen et al., 2007*).

## Discussion

A majority of the components of the core DNA replication initiation machinery appear evolutionary conserved across eukaryotes, suggesting that structural and mechanistic aspects of replicative helicase loading are as well. In this study, we have investigated the architecture of metazoan ORC, with an emphasis on Orc6, which has been thought to play distinct roles during pre-RC assembly in *S. cerevisiae* compared to metazoans. We find that the C-terminal domain of Orc6 plays a critical and highly conserved function in localizing the subunit to the core ORC1–5 subcomplex in flies, humans and budding yeast. The significance of this association is emphasized by the fact that a mutation in this region of Orc6, which has been linked to the onset of Meier-Gorlin syndrome, abrogates formation of fully intact ORC hexamers, suggesting a molecular mechanism for reduced origin licensing in this subset of MGS patients.

### Subunit architecture of metazoan ORC

5 out of the 6 ORC subunits are well conserved between yeast and metazoans, predicting that substantial structural homology should exist between the respective ORC complexes (*Duncker et al., 2009*). However, this paradigm has been challenged by apparent differences in shape between previous *Sc*ORC and *Dm*ORC EM reconstructions; for example, *Sc*ORC forms a somewhat flattened crescent, whereas *Dm*ORC has a pronounced third dimension and adopts a more helical conformation (*Speck et al., 2005*; *Clarey et al., 2006*) (compare *Figure 1—figure supplement 2A*, *Figure 10A*). These differences were too significant to allow the subunit order determined for *Sc*ORC to be mapped directly onto *Dm*ORC (*Chen et al., 2008*; *Sun et al., 2012*), an analysis complicated by the low-resolution nature of initial ORC EM models (*Speck et al., 2005*; *Clarey et al., 2006*, *2008*).

The improved resolution EM structure of *Dm*ORC presented here, together with experimental mapping of the ORC subunits in the EM structure, provides the first organizational model for the architecture of metazoan ORC (*Figure 10B*). Characteristics observed in the current structure, but not in previous ones, include a more featured AAA⁺-ring and likely subunit borders (*Figure 2B*). Importantly, this new EM structure of ATPγS-bound *Dm*ORC is more similar to budding yeast ORC EM structure (*Speck et al., 2005*; *Sun et al., 2012*, *2013*), although the fly complex displays more surface features and is thicker when viewed from the lateral edge of the AAA⁺ core (*Figure 10A*). Moreover, the AAA⁺ subunit order determined here for *Dm*ORC—Orc1-Orc4-Orc5-Orc2–Orc3—agrees with that published for *Sc*ORC (*Figure 10B*). Although both the published budding yeast ORC structures and the current *Dm*ORC structure lack the resolution to unambiguously locate AAA⁺ and winged helix domains of the ORC subunits, they nonetheless demonstrate fundamental similarities in ORC structure from yeast to metazoans.

One notable difference we do observe between yeast ORC and *Dm*ORC is that *Drosophila* Orc6 is flexibly tethered to ORC through Orc3 and is hence invisible in the EM 3D structure (*Figures 3A and 10B*). By comparison, *S. cerevisiae* Orc6 has been reported to contribute more significantly to EM density of yeast ORC (*Chen et al., 2008*) and has been reported to bind to Orc2 and not Orc3 (*Sun et al., 2012*). Our pull-down data suggest that the more stable conformation seen for budding yeast Orc6 may be due to the existence of binding sites for the subunit on both Orc2 and Orc3 within ORC1–5, as opposed to just Orc3 in the *Drosophila* complex (*Figure 9*).

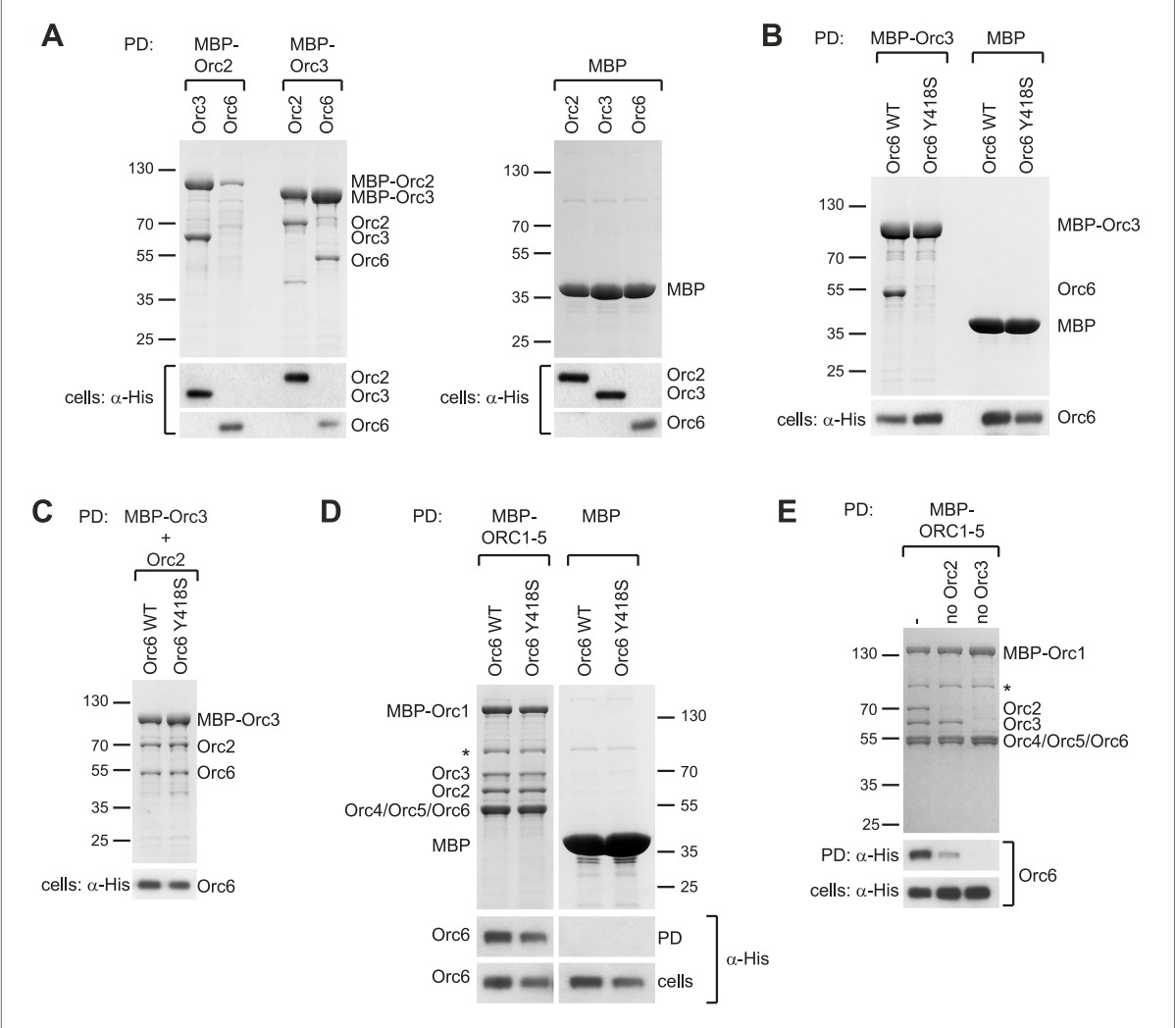

**Figure 9**. The C-terminal domain in *S. cerevisiae* Orc6 interacts with Orc3 and an MGS-like mutation abrogates this interaction. (**A**) *S. cerevisiae* Orc6 binds Orc3. His-tagged Orc2, Orc3 or Orc6 were co-expressed with MBP–Orc2, MBP–Orc3 or MBP only in High5 cells and cell extracts were subjected to pull-downs (PD) using amylose beads. Orc6 and Orc3, as well as Orc2 and Orc3, interact directly with each other. MBP–Orc2 was very unstable when expressed without Orc3. The asterisk indicates a degradation product of either MBP–Orc3 or Orc2. (**B**) An MGS-like mutation in the C-terminal domain of *S. cerevisiae* Orc3 abolishes binding of Orc6 to Orc3. Wild-type (WT) Orc6 or Orc6 containing a mutation corresponding to the MGS mutation in human Orc6 (Y418S) were co-expressed with either MBP–Orc3 or MBP only in insect cells. Only wild-type *S. cerevisiae* Orc6, but not the Y418S mutant, bound to MBP–Orc3 in amylose pull-downs. (**C**) The MGS-like mutation in *S. cerevisiae* Orc6 allows incorporation of Orc6 into the Orc2-Orc3-Orc6 subcomplex. Orc6-WT and Orc6-Y418S were co-expressed with untagged Orc2 and MBP–Orc3 in High5 cells. In amylose pull-downs, a ternary complex containing Orc2, Orc3, and Orc6 was recovered for both Orc6-WT and Orc6-Y418S. Since Orc6-Y418S does not interact with Orc3 alone (panel **B**) this result suggests that a second binding site for Orc6 exists in *S. cerevisiae* Orc2 which is not sensitive to the MGS-like mutation in the C-terminal domain of Orc6. (**D**) The *S. cerevisiae* Orc3–Orc6 interaction is not essential for hexameric ORC formation. *S. cerevisiae* ORC subunits were co-expressed in High5 cells with Orc6-WT or Orc6-Y418S and cell extracts were subjected to pull-downs (PD) using amylose beads binding to MBP–Orc1. All six subunits co-purify with MBP–Orc1. Due to the similar migration of Orc6 and Orc4/Orc5, bead-bound Orc6 was detected by Western blotting using anti-His antibody. No Orc6 was recovered in control pull-downs from cells expressing Orc6 and MBP. (**E**) At least two different binding sites recruit *S. cerevisiae* Orc6 into ORC. *S. cerevisiae* ORC subunits were co-expressed in insect cells to reconstitute ORC containing all six ORC subunits (-), ORC lacking Orc2 (no Orc2) and ORC lacking Orc3 (no Orc3). Amylose pull-downs targeting MBP–Orc1 result in reduced Orc6 incorporation into ORC in the absence of Orc2, and no Orc6 is recovered when Orc3 is absent. Omission of Orc3 also resulted in loss of co-purification of Orc2. As in (**D**) Orc6 was detected by Western blotting (WB). Asterisks in (**D** and **E**) indicate a protein contaminant binding to amylose beads. In (**A**–**E**) Coomassie-stained SDS-PAGE gels are shown for pull-downs and Western blots (WB) of whole cell extract verify that all His-tagged subunits were expressed.

The following figure supplements are available for figure 9:

**Figure supplement 1**. The C-terminus of *S. cerevisiae* Orc6 is essential for yeast growth.

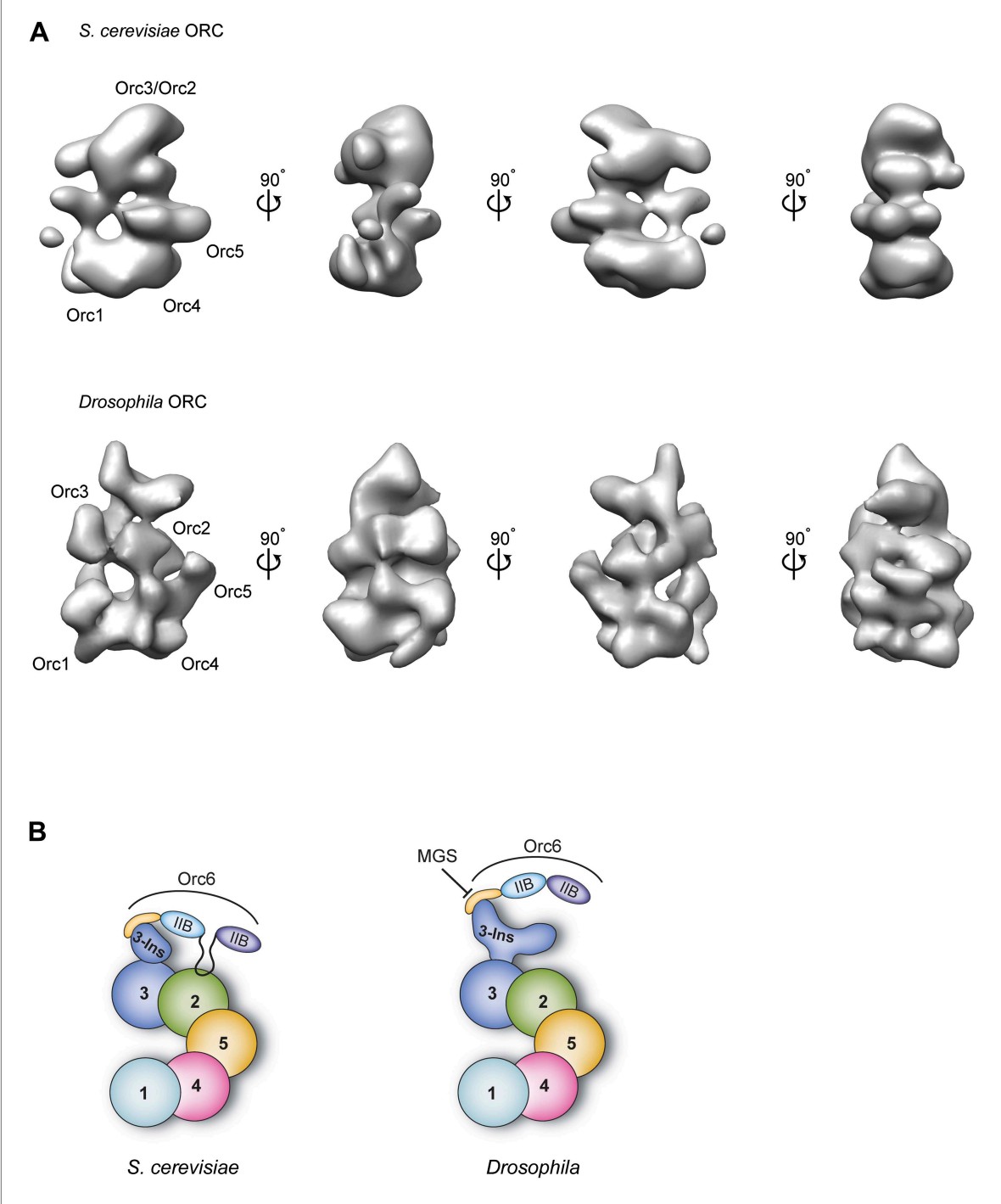

**Figure 10**. Comparison of metazoan and *S. cerevisiae* ORC architectures and model of how the MGS mutation in Orc6 impedes ORC formation. (**A**) Different views of *S. cerevisiae* ORC (EMDB 1156, *Speck et al., 2005*) and the new *Drosophila* ORC structure (this study) are compared (a more recent, higher-resolution EM model of budding yeast ORC in isolation (*Sun et al., 2012*) has been not deposited into the EM database and so could not be used here). The localization of ORC subunits as reported for *S. cerevisiae* ORC (*Chen et al., 2008*; *Sun et al., 2012*) and *Drosophila* ORC (this study) are indicated. (**B**) Model for AAA+ subunit architecture and Orc6 recruitment in *Sc*ORC and *Dm*ORC. In contrast to *Drosophila*, yeast Orc6 also interacts with Orc2 (*Sun et al., 2012*). The MGS mutation in the C-terminal domain of Orc6 impedes binding of Orc6 to the Orc3 insertion and recruitment of Orc6 into metazoan ORC, likely resulting in reduced origin licensing. Numbers indicate the respective ORC subunit, 'Ins' the Orc3 insertion, and 'IIB' the TFIIB domains of Orc6.

## A replicative function for the conserved C-terminal helix in *Drosophila* Orc6

In *Drosophila*, Orc6 is an essential protein, with reduced Orc6 levels resulting in reduced DNA replication in both cell culture and flies (*Chesnokov et al., 2003*; *Balasov et al., 2009*). Replicative functions have been attributed to the N-terminal TFIIB-like region of Orc6, whereas the C-terminal domain is required for cytokinesis and interacts with the septin Pnut (*Chesnokov et al., 2003*; *Balasov et al., 2007*, *2009*; *Huijbregts et al., 2009*). Our results here show that the C-terminus of Orc6 is also critical for DNA replication. Orc6 is a stable component of *Drosophila* ORC, and ORC lacking Orc6 cannot support DNA replication in vitro in ORC-depleted *Drosophila* extracts (*Chesnokov et al., 2001*). Since the C-terminus of Orc6 mediates the association of Orc6 with ORC (*Figures 4–7*), we predict that truncations or mutations abrogating this interaction should impair DNA replication. Indeed, the expression of C-terminally truncated or mutated Orc6 transgenes in *orc6*-null flies results in reduced levels of Mcm4 associated with chromatin (*Figure 7C*), consistent with reduced levels of replicative helicase loading. These observations may explain the previously described cell-cycle arrest in mitosis that coincides with broken and uncondensed chromosomes, as well as reduced BrdU incorporation in neuroblasts observed in these fly strains, suggesting DNA is underreplicated (*Balasov et al., 2009*). Moreover, Orc6 constructs lacking part of the C-terminal domain are compromised for initiating DNA replication in *Drosophila* extracts (*Balasov et al., 2007*). Our results provide a mechanistic explanation for these observations: since Orc6 contributes to recruitment of ORC to chromatin through the binding of its TFIIB-like domains to DNA (*Balasov et al., 2007*), loss of the interaction between Orc6 and ORC1–5 likely reduces the levels of chromatin-associated ORC and impairs ORC-dependent pre-RC assembly. The more moderate replicative phenotype of Orc6 mutations that affect Orc6 recruitment into ORC, compared to Orc6 mutations that affect the DNA binding activity of Orc6 (*Balasov et al. 2007*, *2009*), suggests that Orc6 might make contacts with non-ORC pre-RC components that help recruit ORC to DNA during pre-RC assembly. Since a majority of ORC in fly ovary and embryo extracts is not bound to origins, we could not test if mutant Orc6 is recruited into the pre-RC with our assays.

## A molecular link between an MGS mutation in Orc6 and impaired origin licensing

Orc6 is one of several pre-RC components found mutated in Meier-Gorlin syndrome (MGS) patients. In certain MGS patients, compound heterozygous mutations inactivate one *ORC6* allele and introduce a missense mutation in the other allele that substitutes serine for tyrosine in a conserved C-terminal helix of the Orc6 protein. We show that tyrosine-to-serine mutation reduces the affinity of Orc6 for the core ORC1–5 complex in both *Drosophila* and humans (*Figures 6 and 8*), a finding that would appear to implicate this subunit in the formation of the pre-RC. Interestingly, earlier observations have appeared inconsistent with such a role of Orc6 in some metazoans; for example, in contrast to the *Drosophila* system, Orc6 is only loosely or transiently associated with ORC1–5 in humans as well as in other metazoans, while in vitro DNA replication experiments using *Xenopus* extracts have suggested that Orc6 may not be required for DNA replication in this context (*Chesnokov et al., 1999*; *Dhar and Dutta, 2000*; *Chesnokov et al., 2001*; *Dhar et al., 2001*; *Gillespie et al., 2001*; *Vashee et al., 2001*, *2003*; *Giordano-Coltart et al., 2005*; *Ranjan and Gossen, 2006*; *Siddiqui and Stillman, 2007*). Nevertheless, evidence is emerging that Orc6 is essential for DNA replication in these species as well. First, depletion of Orc6 in human cell lines causes licensing and DNA replication defects (*Prasanth et al., 2002*; *Stiff et al., 2013*). Second, human Orc6 is required in vitro for efficient DNA replication in Orc6-depleted *Xenopus* extracts (*Liu et al., 2011*); the initial discrepancies between the *Drosophila* system and other metazoan systems may reflect different extents of immunodepletion of Orc6 from *Xenopus* extracts due to the use of different antibodies. Third, tethering of human Orc6 to DNA in human cells creates a functional origin and recruits other factors of the licensing machinery (*Thomae et al., 2011*). Finally, immunodepletion of Orc6 from HeLa cell extracts impairs pre-RC assembly on plasmid DNA (*Thomae et al., 2011*).

Our data lend credence to the idea that the MGS mutation identified in human Orc6 is hypomorphic, and that the diminished affinity of MGS mutant Orc6 for ORC1–5 compromises pre-RC formation. Consistent with this interpretation, we observe reduced but weak binding of a *Drosophila* Orc6 construct bearing the MGS mutation to either Orc3 or ORC1–5 in equilibrium binding experiments and pull-down assays (*Figure 6B,D*). Nonetheless, additional interactions of Orc6 with pre-RC components other than ORC subunits during origin licensing could also contribute to Orc6 incorporation into the

pre-RC, explaining why the MGS mutation is not lethal in humans. In addition to its canonical function during replication initiation, metazoan Orc6 is also essential for cytokinesis (*Prasanth et al., 2002*; *Chesnokov et al., 2003*; *Bernal and Venkitaraman, 2011*), helps to control centrosome copy number and, along with other pre-RC components, appears to be essential for formation of primary cilia (*Stiff et al., 2013*). The C-terminal domain of Orc6 harboring the MGS mutation is important for cytokinetic function, both in *Drosophila* and humans (*Chesnokov et al., 2003*; *Bernal and Venkitaraman, 2011*). At this point it is not known whether the MGS mutation specficially impairs Orc6's role in cytokinesis, or whether the C-terminal domain of Orc6 is involved in regulating centrosome copy number and cilia formation in humans. In addition, it is not understood how the defects in origin licensing caused by other MGS mutations would feed back to ciliogenesis (*Stiff et al., 2013*). Nevertheless, given the fact that many independent mutations in different human ORC subunits, where some have been shown to affect replication functions directly, and hypomorphic mutations in the ATR replication check point protein lead to primordial dwarfism, it is hard to escape the inference that loss of ORC replication licensing functions are indeed epistatic to the MGS phenotype. Interestingly, MGS patients do not appear to have an increased predisposition to cancer, suggesting chromosome stability is not compromised (*de Munnik et al., 2012a, b*). How these observations are linked, how they relate to the phenotypes observed in MGS patients, and how specific MGS mutations intercept these processes will be important tasks for future studies.

## Conserved domain organization in *S. cerevisiae* and metazoan Orc6

*S. cerevisiae* Orc6 and metazoan Orc6 have been shown to perform distinct roles in initiating DNA replication: metazoan Orc6 binds DNA through its TFIIB-like domains and helps recruiting ORC to chromatin (*Chesnokov et al., 2001*; *Balasov et al., 2007*; *Liu et al., 2011*), while *S. cerevisiae* Orc6 does not bind DNA but has been shown to be directly essential for the loading step of the MCM double hexamer onto DNA (*Lee and Bell, 1997*; *Balasov et al., 2007*; *Chen et al., 2007*; *Duncker et al., 2009*; *Evrin et al., 2009*; *Remus et al., 2009*; *Gambus et al., 2011*; *Liu et al., 2011*; *Takara and Bell, 2011*; *Fernandez-Cid et al., 2013*; *Frigola et al., 2013*). Up to now, this difference has seemed consistent with the apparent lack of significant sequence homology between these proteins. However, we found that the C-terminal helix that is crucial for Orc6 association with metazoan ORC is actually highly conserved in *S. cerevisiae* Orc6, both in sequence and in function (*Figure 4—figure supplement 1*, *Figure 9*). Moreover, we found that the sequence homology between metazoan and *S. cerevisiae* Orc6 extends to the TFIIB-like folds identified in metazoan Orc6 (*Figure 4—figure supplement 1*). Interestingly, budding yeast Orc6 lacks multiple basic amino acids that are conserved in metazoan Orc6, resulting in a lower isoelectric point for its TFIIB domains (pI = 7.5) compared to the *Drosophila* protein (pI = 9.0). This difference in surface charge may explain the distinct DNA binding propensities of metazoan and *S. cerevisiae* Orc6 (*Lee and Bell, 1997*; *Balasov et al., 2007*; *Chen et al., 2007*; *Liu et al., 2011*). Nonetheless, because the TFIIB domains are preserved across eukaryotic Orc6 orthologs, it is tempting to speculate that these folds also play an important conserved function during origin licensing. While recent reports indicate that Orc6 may not be essential for MCM2–7 recruitment to origins, it is essential for loading of MCM double hexamers onto origin DNA (*Fernandez-Cid et al., 2013*; *Frigola et al., 2013*). The loading function of Orc6 may involve binding of Cdt1 or another pre-RC component through direct protein interactions, as has been reported for *S. cerevisiae* Orc6 and Cdt1 (*Chen et al., 2007*). Notably, of the two regions of Orc6 that were shown to independently interact with Cdt1, each contain one of the TFIIB-like domains identified here in yeast Orc6 (*Chen et al., 2007*), suggesting Cdt1 binding may be conserved in metazoan Orc6 as well. Along these lines, acquisition of the MGS mutation in human Orc6 would be predicted to reduce licensing capacity and MCM loading by impeding Orc6 recruitment into ORC and pre-replicative complex formation, respectively (*Figure 10B*). Future efforts using reconstituted systems will be needed to test these concepts further.

## Materials and methods

### Expression and purification of recombinant ORC

#### Untagged *Drosophila* ORC for EM studies

*Drosophila* ORC was recombinantly expressed in High5 cells using the baculovirus expression system. Individual ORC subunits were cloned into the pFastBac transfer vector (Invitrogen Life Technologies, Grand Island, NY) or a modified transfer vector containing ligation-independent cloning (LIC) sites.

*Drosophila* Orc1 was cloned with an N-terminal 6 × Histidine (6 × His) tag and *Drosophila* Orc4 with an N-terminal maltose binding protein (MBP) tag, both followed by a tobacco etch virus (TEV) protease cleavage site. The remaining subunits, Orc2, Orc3, Orc4, and Orc6, contained no purification tag. All clones were verified by DNA sequencing to not contain missense or nonsense mutations.

Bacmids were generated from transfer vectors in DH10Bac cells and isolated according to Bac-to-Bac protocol (Invitrogen Life Technologies). Bacmid DNA was transfected into Sf9 cells using Cellfectin II (Invitrogen Life Technologies) to generate initial baculovirus stocks. Viruses were amplified for two generations in Sf9 cells to obtain high titer viruses expressing individual ORC subunits.

For purification of recombinant *Drosophila* ORC, 4 l of High5 cells in shaking culture were co-infected with high titer baculoviruses expressing the six different ORC subunits. In the case of ORC1–5, baculovirus expressing Orc6 was omitted. 48 hr post infection, High5 cells were harvested by centrifugation in a Beckman Avanti J-20 centrifuge at 202 × *g* for 30 min. Cells were resuspended in 25 ml lysis buffer (50 mM Tris HCl pH 7.8, 50 mM imidazole pH 7.8, 300 mM KCl, 10% glycerol, 200 μM PMSF, 1 μg/ml leupeptin) per liter High5 cells and lysed by gentle sonication at 4°C. Lysate was clarified by centrifugation in a Sorvall RC-5B superspeed centrifuge at 38,724 × *g* for 1 hr and subsequently precipitated with 0.78 M $(NH_4)_2SO_4$ for 30 min on ice. After an additional centrifugation step at 38,724 × *g* for 1 hr, the supernatant was filtered with a 1.2 μm Acrodisc syringe filter (PALL Life Sciences, Ann Arbor, MI). This clarified lysate was passed over a 5 ml HisTrap HP column (GE Healthcare Biosciences, Pittsburgh, PA), washed with 60 ml lysis buffer, and eluted with a 50–250 mM imidazole gradient in elution buffer (50 mM Tris HCl pH 7.8, 300 mM KCl, 10% glycerol) over 30 ml. Eluted ORC was subsequently bound to a 20 ml amylose column (New England Biolabs, Ipswich, MA), washed with 50 ml wash buffer (50 mM Tris HCl pH 7.8, 300 mM KCl, 10% glycerol), and eluted with 30 ml elution buffer (50 mM Tris HCl pH 7.8, 300 mM KCl, 10% glycerol, 20 mM maltose). ORC containing fractions were pooled and 6 × His- and MBP-tags on Orc1 and Orc4, respectively, were cleaved off overnight with TEV protease at 4°C. This typically resulted in complete cleavage as judged by a shift in migration of Orc4 during SDS-PAGE before and after TEV digest. Untagged ORC was then concentrated to ~ 2 ml using Centriprep-30K concentrators (EMD Millipore, Billerica, MA) and further purified by gel filtration chromatography on a HiPrep 16/60 Sephacryl S-300 HR column (GE Healthcare Biosciences). The elution volume was consistent with the expected size of the ORC hexamer. Purified ORC was aliquoted, flash frozen in liquid nitrogen and stored at −80°C.

### *Drosophila* ORC containing localization tags for EM

All ORC subunits were cloned into LIC converted baculovirus transfer vectors fused to either an N-terminal MBP-tag followed by a TEV cleavage site or a C-terminal msfGFP-tag. For localization of Orc1, a tandem N-terminal 6 × His-MBP-tag or a combination of N-terminal 6 × His- and C-terminal msfGFP-tags were used to maintain the 6 × His-tag on Orc1 for affinity purification. In the case of Orc1 and Orc2, additional N-terminal 6 × His-MBP- and MBP-tagged constructs, respectively, were generated that lack non-conserved amino acid residues, that is amino acid residues 1–449 for Orc1 and amino acid residues 1–265 for Orc2. Bacmids and baculoviruses were generated as described for purification of untagged ORC.

For expression of MBP-tagged ORCs, 4 l of High5 cells were co-infected expressing 6 × His-Orc1, one MBP-tagged subunit and respective untagged subunits. Expression and purification of ORC complexes were performed as described above with the exception of the TEV protease digestion. This step was omitted to keep the MBP-tag on ORC. GFP-tagged ORCs were expressed by co-infection of High5 cells with baculoviruses expressing 6 × His-Orc1, MBP-Orc4, and respective GFP-tagged and untagged ORC subunits. In the case of ORC-Orc4-GFP, the MBP-tag used for purification was on Orc6. Expression and purification of ORC-GFP complexes were performed as described for untagged ORC, including the TEV cleavage step to remove 6 × −His- and MBP-tags from ORC.

### ORC1–5 for biochemistry

In vitro Orc6 binding experiments were performed with recombinant *Drosophila* ORC containing subunits Orc1–Orc5 (referred to as ORC1–5). This complex was expressed by co-infecting 8 l of High5 cells with baculoviruses expressing 6 × His-Orc1, MBP–Orc4 as well as untagged Orc2, Orc3, and Orc5. Purification was performed as described for hexameric ORC except for two modifications: (1) reducing agents were added to purification buffers (1 mM β-mercaptoethanol during Nickel-affinity and 1 mM DTT during amylose affinity and gel filtration chromatography) and (2) the TEV cleavage

step was omitted, leaving the 6 × His- and MBP-tags on Orc1 and Orc4, respectively. Human ORC1–5 was expressed and purified analogously to *Drosophila* ORC1–5.

## Purification of Orc6

*Drosophila* and human Orc6 proteins were expressed in insect cells. Individual genes were cloned into LIC converted baculovirus transfer vectors adding an N-terminal 6 × His-tag. N- and C-terminal truncations of *Drosophila* Orc6 (amino acids 94–257, 147–257, 187–257, 1–200, 1–220, 1–232) were generated by PCR amplification of specified regions and cloned as N-terminal 6 × His-fusions into baculovirus transfer vectors by ligation-independent cloning. Point mutations in Orc6 were obtained by site-directed mutagenesis. All clones were verified by DNA sequencing. Bacmids and baculoviruses were generated as described above.

For purification of all human and *Drosophila* Orc6 proteins, 2 l of High5 cells were infected with baculoviruses expressing respective wild-type or mutant Orc6 proteins. After 48 hr, cells were harvested and lysate was prepared as described in detail above for *Drosophila* ORC. Briefly, cells were resuspended in 50 ml lysis buffer (50 mM Tris HCl pH 7.8, 50 mM imidazole pH 7.8, 600 mM KCl, 10% glycerol, 1 mM β-mercaptoethanol, 200 µM PMSF, 1 µg/ml leupeptin) and lysed by sonication. Lysate was clarified by centrifugation and a 0.78 M $(NH_4)_2SO_4$ precipitation, followed by another centrifugation step. Subsequently, Orc6 was bound to a 5 ml HisTrap HP column (GE Healthcare Biosciences) and washed with 100 ml lysis buffer. Salt was reduced to 300 mM KCl by doing an additional wash with 50 ml buffer (50 mM Tris HCl pH 7.8, 50 mM imidazole pH 7.8, 300 mM KCl, 10% glycerol, 1 mM β-mercaptoethanol). Orc6 was eluted by increasing the imidazole concentration to 250 mM over 30 ml. Fractions containing Orc6 were pooled, concentrated and further purified by gel filtration chromatography on a HiPrep 16/60 Sephacryl S-200 HR column (GE Healthcare Biosciences).

## Electron microscopy and image processing

### Sample preparation and data collection

For negative stain electron microscopy (EM), untagged *Drosophila* ORC was diluted to 30 nM in buffer containing 20 mM Tris HCl pH 7.8, 300 mM KCl, 5 mM $MgCl_2$ and either no nucleotide or 1 mM ADP, ATP, AMPPNP or ATPγS. 4 µl of sample were adsorbed for 20 s to a glow discharged 400 mesh EM grid containing a continuous carbon film. Grids were stained with 2% uranly formate (wt/vol), slightly blotted from the side to remove excess stain and allowed to dry.

ORC was imaged in a Tecnai T12 BIOTWIN transmission electron microscope equipped with a $LaB_6$ cathode and operated at 120 keV. Images were collected at a nominal magnification of 49,000 (2.18 Å/pixel at the specimen level) with an electron dose of 20–30e$^-$/Å$^2$ using a 4K × 4K TVIPS TemCam-F416 CCD camera and the Leginon software package (*Suloway et al., 2005*).

### Image processing

Initial image processing steps up to the point of particle stack creation were performed in the Appion image processing environment (*Lander et al., 2009*). Initially, power spectra of micrographs were visually inspected and micrographs showing evidence of drift or significant astigmatism were excluded from further processing. CTF parameters of the micrographs were estimated using CTFFIND (*Mindell and Grigorieff, 2003*). Defocus values ranged from −0.4 to −1.2 µm. CTF correction included flipping of micrograph phases using SPIDER (*Frank et al., 1996*). Particles were automatically selected in a reference-free manner using a Difference of Gaussian (DoG) image transform particle picker (*Voss et al., 2009*). Particle stacks were extracted with an initial box size of 160 × 160 pixel and subsequently binned by a factor of two, resulting in a pixel size of 4.36 Å/pixel. Particle images were normalized using the Ramp method in XMIPP, which included the removal of pixels with values 4.5σ above or below the mean (*Sorzano et al., 2004*; *Scheres et al., 2008*). Image outliers containing particle aggregates or precipitates were identified by sorting particle images within the stack based on general statistics parameters using the sort-by-statistics function in XMIPP and removed from the image stack. The resulting cleaned image stacks contained 35,000 particles for untagged *Drosophila* ORC in the presence of ATPγS and between 20,000 to 40,000 particles for data sets of ORC complexes in the presence of other nucleotides and for ORC containing EM localization tags in individual subunits.

2D image analysis was performed by iterative alignment and classification of the particle images using the refine2d.py function in the EMAN software package, generating between 100–200 classes for each complex (*Ludtke et al., 1999*). Class averages of untagged ORC indicated that ORC adopts

a slightly preferred orientation on the carbon film. To increase the number of orientations for 3D reconstruction, we recorded additional micrographs at 30° nominal stage tilt. Particles were automatically selected with DoG picker (*Voss et al., 2009*) and extracted using a 256 × 256 box size. CTF parameters of micrographs as well as tilt axes and tilt angles were estimated with CTFTILT and local defocus values for each particle were calculated based on the particle position in the micrograph and the CTFTILT estimates (*Mindell and Grigorieff, 2003*). Phases of particle images were corrected with SPIDER. After CTF correction, the box size was reduced to 160 × 160 pixel and particle images were binned by a factor of two, resulting in an image stack with a final box size of 80 × 80 (4.36 Å/pixel) containing 35,000 particles. This stack was combined with the 0° tilt data, yielding an image stack containing 70,000 particles which was used for further processing.

3D volumes of ORC were reconstructed by iterative projection-matching refinement using SPIDER (*Frank et al., 1996*). As a starting model, we used the previous 34 Å-resolution EM reconstruction of *Drosophila* ORC (EMDB-1252) (*Clarey et al., 2006*). To avoid model bias, the starting volume was low-pass filtered using a Butterworth filter with stop band and pass band of 100 Å and 200 Å, respectively. Images of untilted and 30° tilted particles were assigned Euler angles by comparison with reference projections and back-projected to obtain a new volume, which served as a reference during the next refinement iteration. The initial angular increment for projection-matching was 30° and was decreased progressively to 8° during the refinement. The data converged very quickly to a stable 3D volume of ORC with a resolution of 22 Å based on the 0.5 Fourier shell correlation (FSC) criterion.

3D reconstructions of MBP- or GFP-tagged ORC were performed using the processing scheme above. The new, untagged ORC reconstruction low-pass filtered to ~50 Å with a Butterworth filter served as an initial starting model.

## Freehand test

The absolute hand of *Drosophila* ORC was determined using the Freehand test (*Rosenthal and Henderson, 2003*; *Henderson et al., 2011*). Tilt-pairs of untagged *Drosophila* ORC particles recorded at 0° and 50° nominal stage tilt were manually picked with XMIPP. 67 tilt-pairs were subjected to the fast-freehand test as implemented by John Rubinstein (Hospital for Sick Children, Toronto, Canada) and adapted by Michael Cianfrocco (University of California Berkeley, Berkeley) to use the EMAN2 and SPARX libraries for alignment of particles to model projections (*Hohn et al., 2007*; *Tang et al., 2007*). The agreement between the tilted particles and the model projections is described by a correlation coefficient that is plotted as a function of tilt axis and tilt angle. For ORC, a correlation peak was found at a tilt angle of −50° instead of +50°, suggesting the hand of the calculated ORC EM volume was incorrect. For further interpretation of the ORC EM volume, the hand was inverted. As a control, an analogous freehand test was performed with the ribosome, the absolute hand of which is known. In the case of the ribosome, peak correlation was observed at the correct tilt angle and not the inverse, as is expected for the correct absolute hand. The ORC EM volume was deposited in the EM data bank (accession number EMD-2479).

## In vitro binding experiments of Orc6 to ORC1–5

### In vitro pull-downs

To probe the interaction of *Drosophila* Orc6 with *Drosophila* ORC1–5, 100 pmol of the respective purified Orc6 protein and 100 pmol of purified ORC1–5 (or 100 pmol Orc6 alone in control reactions) were mixed in a total volume of 50 µl (final reaction buffer contains 50 mM Tris HCl pH 7.8, 300 mM KCl, 10% glycerol, 1 mM DTT) and incubated on ice for 1 hr. Reactions were added to 25 µl amylose beads (New England Biolabs) for 30 min to bind ORC1–5 via MBP–Orc4. Amylose beads were washed three times with 1 ml of reaction buffer and bound proteins were eluted from beads with 20 µl reaction buffer containing 20 mM maltose. Eluted proteins were analyzed by SDS-Page and stained with Coomassie Brilliant Blue.

To test binding of wild-type and MGS mutant human Orc6 to human ORC1–5, 100 pmol ORC1–5 were incubated with either 100 pmol or 1 nmol of the respective Orc6 protein. Otherwise, conditions used for binding reactions and pull-downs were identical to those described for *Drosophila* proteins.

### Fluorescence anisotropy

To measure the effect of mutations or truncations of Orc6 on the affinity of Orc6 for ORC1–5, fluorescence anisotropy experiments were performed using *Drosophila* and human Orc6 proteins (wild type,

truncations and point mutants), N-terminally labeled with Alexa Fluor 488 5-SDP ester (Invitrogen Life Technologies). For labeling, 200 µl of the respective Orc6 protein at ~10 mg/ml were dialyzed overnight into buffer containing 50 mM HEPES pH 7.5, 300 mM KCl, 10% glycerol, 1 mM DTT. Protein was then mixed with 0.5 mg of Alexa Fluor 488 5-SDP ester and incubated 1 hr at 4°C. The labeling reaction was stopped by the addition of 30 µl 1M lysine in buffer 50 mM Tris HCl pH 7.8, 300 mM KCl, 10% glycerol, 1 mM DTT. Free label was removed by buffer exchanging protein into 50 mM Tris HCl pH 7.8, 300 mM KCl, 10% glycerol, 1 mM DTT using Micro Bio Spin six columns (Biorad, Hercules, CA). Protein and dye absorbance were measured at 280 nm and 495 nm, respectively, and corrected protein and incorporated dye concentrations were calculated according to the manufacturer's instructions. Labeling efficiencies ranged from ~10% to ~50%.

Binding reactions were performed in 80 µl reactions containing twofold serially diluted ORC1–5 ranging from 1 µM to 122 pM and 30 nM Alexa Fluor 488 labeled Orc6 in binding buffer (50 mM Tris HCl pH 7.8, 300 mM KCl, 5% glycerol, 1 mM DTT, 0.1 mg/ml BSA). After incubation on ice for 1 hr, 20 µl of each reaction were transferred into a 384-well plate (Corning Life Sciences, Tewksbury, MA) in triplicates and FP was measured in a Victor 3.5 multilabel plate reader (Perkin Elmer, Waltham, MA) with excitation and emission filters of 485 and 535 nm, respectively. Measured fluorescence polarization values were converted into anisotropy (FA) values and subsequently background-subtracted using a no-ORC1–5 control reaction (yields ΔFA). Averages and standard deviations were calculated from three independent binding experiments and plotted as a function of ORC1–5 concentration. Data were fit to the explicit solution of the single-site binding equation by nonlinear regression analysis (*Heyduk and Lee, 1990*; *Stein et al., 2001*).

## Pull-downs from insect cells

To probe interactions of ORC subunits in vivo when co-expressed in insect cells, 50 ml of High5 cells were co-infected with baculovirus expressing respective subunits. Bait subunits were tagged N-terminally with MBP and prey subunits tested for interaction with the bait were tagged with a 6 × His tag at the N-terminus. This allowed us to ensure expression of the prey protein in whole cell extract by Western blotting. 2 days post infection, High5 cells were harvested by centrifugation at 1,000 × *g* and were resuspended in 1.4 ml lysis buffer (50 mM Tris HCl pH 7.8, 300 mM KCl, 10% glycerol, 1 mM DTT, 200 µM PMSF, 1 µg/ml leupeptin). Cells were lysed by gentle sonication for 10 s and subsequently centrifuged at ~16,000 × *g* for 30 min ($NH_4$)$_2SO_4$ was added to a final concentration of 0.78 M to the lysate and samples were incubated on ice for 30 min. After an additional centrifugation at ~16,000 × *g* for 30 min, the clarified lysate was added to 50 µl of Amylose beads (New England Biolabs) to bind MBP-tagged bait subunit. After 30 min incubation, beads were washed three times with 1 ml of lysis buffer. Bound proteins were eluted either by addition of 40 µl of 20 mM maltose in lysis buffer or by 40 µl of SDS loading dye. Eluted proteins were analyzed by SDS-PAGE and subsequent staining with Coomassie Brilliant Blue.

To confirm that 6 × His-tagged prey proteins are expressed, whole cell extracts were analyzed by Western blotting. 100 µl of High5 cells were harvested and resuspended in 100 µl of SDS-loading buffer. Samples were boiled and 5 µl were loaded on an SDS-PAGE gel. After electrophoresis, proteins were transferred to Immobilon-P$^{SQ}$ membrane (EMD Millipore) using a Trans-Blot semi-dry electrophoretic transfer cell (Biorad). Membranes were probed for His-tagged proteins using a monoclonal anti-His antibody (GE Healthcare Biosciences) at a dilution of 1:10,000 in PBS-T containing 5% (wt/vol) nonfat milk. HRP-conjugated goat anti-mouse IgG (Pierce Thermo Fisher Scientific, Rockford, IL) was diluted 1:20,000 in PBS-T and used as a secondary antibody.

## Immunoprecipitation of ORC from fly ovaries

Heterozygous flies *orc6$^{35}$/Cy, GFP-orc6-wt*; *orc6$^{35}$/Cy, GFP-orc6-W228A/K229A*; *orc6$^{35}$/Cy, GFP-orc6-D224A/Y225A* contain endogenous *orc6* deletion and GFP–Orc6 fusion transposons in which designated amino acids were mutated to alanine. 10–12 freshly dissected ovaries were crushed with a glass homogenizer in 100 µl of high salt IP buffer (25 mM HEPES pH 7.6, 12.5 mM $MgCl_2$, 100 mM KCl, 0.1 mM EDTA, 450 mM NaCl, 0.01% Triton X100) and extracted for 1 hr at 4°C with continuous rotation. Extracts were pre-cleared by centrifugation at 15,000 × *g* for 15 min. Supernatant was diluted three times with regular IP buffer (25 mM HEPES 7.6, 12.5 mM $MgCl_2$, 100 mM KCl, 0.1 mM EDTA, 0.01% Triton X100). Protein A-Sepharose (BioVision, Milpitas, CA) and rabbit polyclonal Orc2 antibodies were incubated with supernatant for 3 hr, washed three times

with IP buffer (20–30 min) and diluted in 10 µl of IP buffer. Samples were boiled in loading buffer, separated in 10% SDS-polyacrylamide gels and transferred to Immobilon-P membrane (EMD Millipore). ORC subunits were detected by Western blotting with anti-GFP monoclonal antibody (Clontech Laboratories, Mountain View, CA) as well as anti-Orc3 or anti-Orc5 antibodies. In addition, GFP fluorescence of samples was directly measured immediately after washing. 100 µl were transferred into a 384-well plate and GFP fluorescence was detected on Multi-Mode Microplate Reader Synergy 2 (BioTek, Winooski, VT) with excitation filter 485/20, emission filter 528/20 and sensitivity 110. Data were normalized to $orc6^{35}/Cy$ strain fluorescence emission intensity.

## MCM2–7 chromatin association in *Drosophila* third instar larvae brains

Homozygous ($orc6^{35}/orc6^{35}$, GFP-orc6-wt; $orc6^{35}/orc6^{35}$, GFP-orc6-W228A/K229A; $orc6^{35}/orc6^{35}$, GFP-orc6-D224A/Y225A) or heterozygous ($orc6^{35}/Cy$, GFP-orc6-wt; $orc6^{35}/Cy$, GFP-orc6-W228A/K229A; $orc6^{35}/Cy$, GFP-orc6-D224A/Y225A) third instar larvae were selected based on fluorescence of the *Cy*-YFP balancer. Brains were dissected and imaginal discs removed. Soluble fraction of MCM2–7 was extracted with 450 mM NaCl, 0.5% NP-40 in PBS buffer for 1 hr. Insoluble proteins were pelleted, the pellet washed three times, and pellets from 8–10 brains per lane were loaded onto an SDS-PAGE gel and Mcm4 detected by Western blotting with Mcm4 rabbit polyclonal antibodies.

## Yeast Orc6 complementation analysis

Orc6 was expressed in yeast under the control of its own promoter using plasmid pSPB66 (a kind gift from SP Bell). Point mutations and premature stop codons were introduced by site-directed mutagenesis and verified by DNA sequencing. The yeast Orc6 degron strain ySC166 (*Chen et al., 2007*) was transformed with pSPB66 or mutant derivatives and transformants were selected for the plasmid in permissive growth conditions ($Cu^{2+}$, dextrose, 25°C). Subsequently, transformants were replica-plated and growth assessed in non-permissve conditions (no $Cu^{2+}$, galactose, 37°C).

## Multiple sequence alignment of Orc3 and Orc6

Multiple protein sequence alignments were performed with MAFFT (*Katoh et al., 2005*; *Katoh and Toh, 2008*). Accession numbers used are as follows. Orc3 alignment with other eukaryotic ORC subunits and archaeal Orc1/Cdc6: *Sulfolobus solfataricus* Orc1–1 (PDB code 2qby, chain A), *Sulfolobus solfataricus* Orc1–2 (AAK41068), *Sulfolobus solfataricus* Orc1–3 (PDB code 2qby, chain B), *Aeropyrum pernix* Orc1 (PDB code 2v1u, chain A), *A. pernix* Orc2 (PDB code 1w5s), *Pyrobaculum aerophilum* Cdc6 (PDB code 1fnn; AAL62992), *Pyrococcus furiousus* Orc1 (AAL80141), *Archeoglobus fulgidus* Cdc6 (AAB90989), *Drosophila melanogaster* Orc1 (AAF59236), *D. melanogaster* Orc2 (AAF55006), *D. melanogaster* Orc4 (AAF47276), *D. melanogaster* Orc5 (AAC46956), *S. cerevisiae* Orc3 (AAB38249), *C. albicans* Orc3 (EAK93535), *S. pombe* Orc3 (AAF05949), *N. crassa* Orc3 (EAA36220), *Arabidopsis thaliana* Orc3 (AAT37463), *Oryza sativa* Orc3 (BAC56110), *Homo sapiens* Orc3 (AAT38109), *Mus musculus* Orc3 (NP_056639), *X. tropicalis* Orc3 (AAI35907), *Danio rerio* Orc3 (AAH45352), *D. melanogaster* Orc3 (AAF58411), *Aedes aegypti* Orc3 (EAT44270).

Alignment of Orc6 protein sequences: *S. cerevisiae* (AAA21822), *C. albicans* (EAK99180), *S. pombe* (NP_596222), *A. fumigatus* (EDP49624), *N. crassa* (CAD37038), *A. thaliana* (AEE30748), *O. sativa* (EAZ40750), maize (ACG41692), soybean (ACU17985), *H. sapiens* (AAD32666), *M. musculus* (AAD32667), *Rattus norvegicus* (AAI01872), *X. tropicalis* (AAI35542), *D. rerio* (AAH56528), *Branchiostoma* floridae (EEN60699), *Saccoglossus kowalevskii* (XP_002740148), *D. melanogaster* (AAF58890),*A. aegypti* (EAT37789), *Bombyx mori* (ADD10142), *Trichoplax adhaerens* (EDV21277), *Phytophthora infestans* (EEY55501), *Dictyostelium discoideum* (EAL65038), *Polysphondylium pallidum* (EFA84700).

## Acknowledgements

We thank Michael Cianfrocco for help with the Freehand test, Ann Fisher (Cell Culture Facility, UC Berkeley) for assistance with insect cell growth, and David King (HHMI Mass Spectrometry Laboratory, UC Berkeley) for mass spectrometric analysis of human Orc6 proteins. We also thank SP Bell for providing yeast strain ySC166 and the yeast wild type Orc6 expression plasmid pSPB66. EN is a Howard Hughes Medical Institute Investigator.

## Additional information

### Competing interests

MRB: Reviewing editor, *eLife*. The other authors declare that no competing interests exist.

### Funding

| Funder | Grant reference number | Author |
|---|---|---|
| National Institutes of Health | GM071747 | James M Berger |
| Howard Hughes Medical Institute | | Eva Nogales |
| Miller Institute for Basic Research in Science | | Franziska Bleichert |
| National Institutes of Health | R37 30490 | Michael R Botchan |
| National Institutes of Health | GM097052 | Igor Chesnokov |

The funders had no role in study design, data collection and interpretation, or the decision to submit the work for publication.

### Author contributions

FB, Conception and design, Acquisition of data, Analysis and interpretation of data, Drafting or revising the article; MB, Conception and design, Acquisition of data, Analysis and interpretation of data; IC, EN, MRB, JMB, Conception and design, Analysis and interpretation of data, Drafting or revising the article

## Additional files

### Major datasets

The following dataset was generated:

| Author(s) | Year | Dataset title | Dataset ID and/or URL | Database, license, and accessibility information |
|---|---|---|---|---|
| Bleichert F, Balasov M, Chesnokov I, Nogales E, Botchan MR, Berger JM | 2013 | Electron microscopy structure of the Drosophila origin recognition complex | EMD-2479; http://www.ebi.ac.uk/pdbe/entry/EMD-2479 | Publicly available at the Electron Microscopy Data Bank (http://www.ebi.ac.uk/pdbe/emdb/). |

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
