## [Decision Letter]

Thank you for sending your work entitled “A Meier-Gorlin syndrome mutation in a conserved C-terminal helix of Orc6 impedes origin recognition complex formation” for consideration at *eLife*. Your article has been favorably evaluated by a Senior editor and 3 reviewers, one of whom is a member of our Board of Reviewing Editors.

The Reviewing editor and the other reviewers discussed their comments before we reached this decision, and the Reviewing editor has assembled the following comments to help you prepare a revised submission.

This manuscript describes the structural and molecular analysis of ORC function in *Drosophila*. The initial portion of the paper derives a higher resolution EM structure of *Drosophila* ORC and map the location of a number of ORC subunits on this structure. This structure helps to resolve previous observed differences between the yeast and *Drosophila* EM structure of ORC bringing helpful agreement between the structures. Using protein sequence analysis, the authors suggest that Orc6 has a similar structure across all eukaryotes (previously Orc6 from fungi was thought to be an outlier). More importantly, the authors find that the C-terminal domain of Orc6 binds to a Orc3 via an insertion between canonical AAA^+^ motifs that is specific to Orc3. Finally, the authors show that a previously identified Meier-Gorlin mutation in Orc6 interferes with this interaction using fly, human, or yeast proteins.

The strength of this manuscript is the identification of the primary interaction site between Orc6 and Orc1-5 and the observation that this is conserved in many (if not all) eukaryotic cells. The EM studies also help to bring together the structural studies of ORC performed in yeast and *Drosophila* cells, although they do not provide any major new insights. One significant weakness is the authors fail to clearly delineate what the defect is when they interfere with the Orc1-5/Orc6 interaction. There is no data in the paper addressing this issue and previous data from one of the authors assessing the function of a mutant that interferes with this interaction was assessed to cause defects in mitosis rather than replication. Despite this, the authors state strongly that the defect is in DNA replication but provide no data to support this conclusion (e.g., measure Mcm2-7 chromatin association). The authors point to reduced BrdU incorporation observed for the mutants in the previous studies as evidence in favor of this interpretation but do not explain why their previous reasoning focused on the arrest point was incorrect. One concern with these analyses overall is that they are looking at terminal phenotypes that could reflect defects in both activities since it is clear that Orc6 participates in both events. Given the data in hand, it would make more sense to be a little less focused on a replication defect.

When the authors submit a revised manuscript, they must address the following points:

1) They should provide further evidence for the molecular defect caused when binding of Orc6 to the rest of the complex is disrupted, or at the very least discuss the ambiguities remaining from their current analysis.

2) In Figure 7, the two mutants of Orc6 are expressed at lower levels than wild type (see input lanes), undermining the conclusion that they are defective for interaction with the ORC complex.

3) Why did the authors make a double mutation (D224A/Y225A) for the functional analysis in flies (Table 1)? This is now no longer directly comparable to the MGS mutation.

4) The MGS mutation (Y418S) should be tested for viability in budding yeast.

5) The previous EM analysis of the *Drosophila* ORC complex reports two structures described as APO and ATP; however the latter was determined in the presence of only 5 μM ATPγS. Considering that the concentration of ATP in the cell is in the low mM range, one can assume that the structures (which are indeed very similar) are both in the APO form. The present structure can be safely considered to be the actual ATP-bound form. Is there anything to be learned regarding conformational changes upon ATP binding from comparing the two, despite the fact that the old structure is at lower resolution?

6) No fitting of the archaeal crystal structure to the EM map is described. Has there been any attempt to do it, considering the potential help provided by the localization of the N/C-terminal tags?

---

## [Author Response]

*1) They should provide further evidence for the molecular defect caused when binding of Orc6 to the rest of the complex is disrupted, or at the very least discuss the ambiguities remaining from their current analysis*.

We have performed additional experiments to address this point of the reviewers and included the results in the main manuscript as Figure 7. Specifically, we have examined the association of Mcm4 with chromatin in larval brains expressing either GFP-Orc6 wild type or GFP-Orc6 mutant transgenes. Our results show that the amount of Mcm4 in the insoluble pellet fraction after salt extraction is greatly reduced in *orc6*-null larvae when mutant, but not wild-type Orc6 transgenes, are expressed. These findings suggest that Orc6 mutants that fail to incorporate into ORC also reduce the amount of MCM2-7 complex loaded onto chromatin, supporting our previous conclusion that Orc6 C-terminal mutants impair DNA replication initiation.

*2) In*
Figure 7*, the two mutants of Orc6 are expressed at lower levels than wild type (see input lanes), undermining the conclusion that they are defective for interaction with the ORC complex*.

We acknowledge the reviewers’ concern that the mutants may be expressed at lower levels than the wild type Orc6. Unfortunately, it is very difficult to adjust the levels in ovary IPs, as “mutant ovaries” tend to be smaller in size (likely due to under-replication). We note that the experiment was done with the same number of ovaries per IP per strain. We have repeated the experiment several times; in every case, mutant GFP-Orc6s do not associate with the rest of the complex.

*3) Why did the authors make a double mutation (D224A/Y225A) for the functional analysis in flies (Table 1)? This is now no longer directly comparable to the MGS mutation*.

We used the double mutants of Orc6 (D224A/Y225A and W228A/K229A) for our experiments because these mutant fly strains were already at hand from previous studies, and because the phenotypes of these strains had been characterized in detail (3). While only the results for the W228A/K229A mutant were published at that time, the experiments with the D224A/Y225A mutant fly strain were carried out in parallel. We agree that these mutations are not directly comparable to the MGS mutation in humans; however, the main conclusion we draw from these mutants is that conserved residues in the C-terminal region of Orc6 are necessary for Orc6 incorporation into ORC and, as we have now shown, for MCM chromatin loading. We are still in the process of making the Orc6 MGS mutant in *Drosophila* and don’t have the results concerning its phenotype yet. Since we have now shown that the C-terminus of Orc6 stimulates loading of MCMs onto chromatin in *Drosophila* and is required for viability in budding yeast (see below), we don’t feel that the MGS mutant fly is critical for our main conclusions.

*4) The MGS mutation (Y418S) should be tested for viability in budding yeast*.

We have now tested Orc6 (Y418S) as well as several Orc6 constructs lacking amino acid stretches from the C-terminus for complementation in yeast (Figure 9—figure supplement 1). In this experiment, we used a strain in which the chromosomal copy of Orc6 has been fused to an N-terminal degron (a kind gift from SP Bell) (10). Under non-permissive growth conditions, we observe that wild-type Orc6 expressed from a centromeric plasmid under the control of its own promoter can complement depletion of the endogenous Orc6 protein and that the yeast are viable. Yeast expressing Orc6Y418S grow similarly to Orc6WT cells, suggesting that this point mutation is not sufficient to disrupt the replicative function of yeast Orc6 in our experimental conditions. This result is consistent with our observation (and that of the Stillman lab (74)) that Orc6 is also recruited into ORC through a yeast-specific interaction with Orc2, a contact that we suggest derives from a pair of domain insertions (one in each protein) that are unique to budding-yeast (Figure 9) (74). By contrast, yeast Orc6 lacking as little as 11 amino acids from its C-terminus is not able to complement endogenous Orc6 depletion, and these cells have a severe slow growth phenotype, a finding consistent with prior published studies from the Bell lab showing that a larger deletion from the C-terminus of *S. cerevisiae* Orc6 is essential for viability (9). Since only replicative functions have been found thus far for yeast Orc6, this result strengthens the proposition that the C-terminus of Orc6, and the patch of conserved homology we have called attention to in this paper, is important for initiation.

*5) The previous EM analysis of the* Drosophila *ORC complex reports two structures described as APO and ATP; however the latter was determined in the presence of only 5 μM ATPγS. Considering that the concentration of ATP in the cell is in the low mM range, one can assume that the structures (which are indeed very similar) are both in the APO form. The present structure can be safely considered to be the actual ATP-bound form. Is there anything to be learned regarding conformational changes upon ATP binding from comparing the two, despite the fact that the old structure is at lower resolution*?

As mentioned in the original manuscript, ATPγS binding appears to trap ORC in a specific conformational state, likely by stabilizing interactions between AAA^+^ subunits Orc1, Orc4, and Orc5. This is also evident in class averages in Figure 1, which show that the core of ORC is much better defined in the presence of ATP?S. Unfortunately, the resolution of our EM reconstruction is, at 22Å, too low to really pinpoint the nature of the conformational changes in individual ORC subunit (see also response to point #6 below).

*6) No fitting of the archaeal crystal structure to the EM map is described. Has there been any attempt to do it, considering the potential help provided by the localization of the N/C-terminal tags*?

We have attempted to fit crystal structures of archaeal Orc1/Cdc6 into our EM volume. However, at our current resolution (22Å) the docking results are not unambiguous. This uncertainty is likely due to conformational changes in ORC subunits with respect to the archaeal Orc1/Cdc6 crystal structures (which themselves have been imaged in a variety of conformational states), whereby different subunits adopting different conformations. Along these lines, the flexible nature of the MBP- and GFP-tags, the modest resolution of the EM reconstruction, and the close spatial proximity of the N- and C-termini of archaeal Orc1/Cdc6 (as seen in available crystal structures) preclude us from pinpointing the N-and C-termini of the subunits in the EM volume. We prefer not to include the fitting into the manuscript because, at this point, we cannot be sufficiently confident whether the conformational changes that we impose on archaeal Orc1/Cdc6 crystal structure during fitting truly reflect the domain organization within eukaryotic ORC subunits. Unambiguous fitting will have to await higher resolution EM structures and/or crystal structures of eukaryotic ORC subunits.